# Kinaema: A recurrent sequence model for memory and pose in motion

**Mert Bulent Sariyildiz    Philippe Weinzaepfel    Guillaume Bono**
**Gianluca Monaci    Christian Wolf**
NAVER LABS Europe
{bulent.sariyildiz|firstname.lastname}@naverlabs.com
https://europe.naverlabs.com/kinaema

## Abstract

One key aspect of spatially aware robots is the ability to "*find their bearings*", *i.e.* to correctly situate themselves in previously seen spaces. In this work, we focus on this particular scenario of continuous robotics operations, where information observed before an actual episode start is exploited to optimize efficiency. We introduce a new model, *Kinaema*, and agent, capable of integrating a stream of visual observations while moving in a potentially large scene, and upon request, processing a query image and predicting the relative position of the shown space with respect to its current position. Our model does not explicitly store an observation history, therefore does not have hard constraints on context length. It maintains an implicit latent memory, which is updated by a transformer in a recurrent way, compressing the history of sensor readings into a compact representation. We evaluate the impact of this model in a new downstream task we call "*Mem-Nav*". We show that our large-capacity recurrent model maintains a useful representation of the scene, navigates to goals observed before the actual episode start, and is computationally efficient, in particular compared to classical transformers with attention over an observation history.

## 1   Introduction

The majority of work in embodied AI, in particular methods based on machine learning, work in *episodic* settings: the agent begins with a clean empty internal representation at every start, dealing with every episode as if it was the first one after unpacking the robot after its purchase. This is in stark contrast to realistic robot operations, where we would expect a robot to be able to exploit information on the scene observed previously.

In this work, we propose a model capable of spatially situating previously observed spaces. While applicable to a broader class of downstream tasks, we focus on navigation and a new continuous variant of the *ImageNav* task: an agent is given a goal image and is required to navigate to the position shown in this image. In the case when the goal is not seen from the starting position, classical solu-

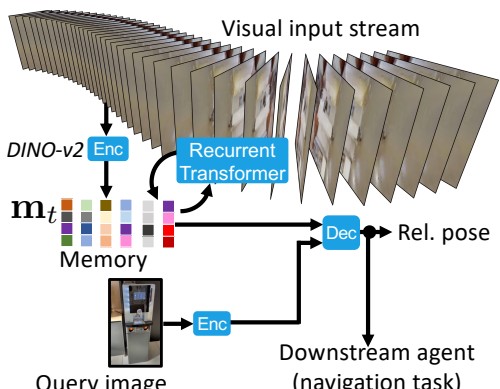

Figure 1: We introduce *Kinaema*, a model capable of situating previously observed spaces: a *recurrent* transformer compresses observed sequences into latent memory and estimates rel. pose of a goal image w.r.t. to its *current* state.

tions resort to an exploration strategy, patrolling the scene until the goal is observed. In our new

39th Conference on Neural Information Processing Systems (NeurIPS 2025).

setting focusing on continuous operation, "*Mem-Nav*", the agent can explore the scene *before* each episode starts to build up a latent representation, but it does not have access to the future goal at this point — it has to build a general representation suitable for *any* potential future goal.

Previous work addressing continuous navigation mainly focuses on map-based solutions [40, 61], building, maintaining and querying a metric or topological map during operation. We address this problem in a purely data-driven way with models and agents which compress a potentially long sequence of visual observations into a high-capacity latent representations, denoted $\mathbf{m}_t$ in Fig. 1. The recurrent nature of our model is a key design feature: given a history of observations of length $N$, sequence models based on transformers update their representations with $O(1)$, essentially stowing away the input, but query it in $O(N^2)$ due to the quadratic complexity of attention. In contrast, our recurrent model, both, updates and queries memory in $O(1)$.[1] Classical recurrent networks, on the other hand, suffer from scaling limitations as their network capacity scales quadratically with respect to memory size [28]. We introduce a new recurrent sequence model, which decouples these aspects and maintains memory in the form of a set of embeddings, which are updated with a transformer.

The main skill required for *Mem-Nav* is the capacity to situate previously observed space, which we directly supervise during a pre-training phase: we train the recurrent model to estimate relative pose between a query/goal image and the current agent position, assuming that the query image depicts a point of the scene which had been previously observed, albeit potentially from a different viewpoint. This text task may be somewhat reminiscent of classical relative pose estimation, but is fundamentally different: compared to classical binocular geometric foundation models comparing pairs of images, eg. *DEBiT* [9], our model compares a single query image to latent agent memory. We combine this pre-training task with a memory based variant of masked-image modeling.

In summary, we introduce the following contributions: (i) a new recurrent sequence model *"Kinaema"* with distributed memory and transformer-based update; (ii) *"Mem-RPE"*, a new task requiring the estimation of relative pose between an image and agent memory; (iii) the integration of the sequence model in a navigation agent trained with Reinforcement Learning; (iv) a new downstream navigation task *"Mem-Nav"* allowing an agent to access observations collected before the episode start.

## 2   Related work

**Visual navigation** has been addressed in robotics for a long time by explicit models [13, 38, 39] based on mapping and localization [12, 34, 56], and explicit planning [32, 52]. ML-based solutions are typically trained on photorealistic simulators [31, 48]. Modular agents [14] decompose the problem in sub-modules, whereas end-to-end trained models directly map input to actions with Reinforcement Learning (RL) [27, 42, 57, 68], Imitation Learning (IL) [19], or offline-RL [54]. **Image goal navigation**, *"ImageNav"*, adds a skill linked to relative pose estimation, which explicit methods have addressed with local feature matching [33], or by retrieving features from a topological map [7]. End-to-end trained agents compare images by extracting binocular features with [1, 53, 67, 69], potentially directly pre-training for RPE [9]. Modular approaches have also been proposed [18, 65].

**Continuous navigation** has previously been cast as problem where episodes are divided in multiple sub-episodes, where later sub-episodes are supposed to exploit information seen earlier. Common formulations are the *K-items scenario* [6], *Multi-Object Navigation* [40, 61] or the *Goat-Bench* [30]. Our *"Mem-Nav"* task decomposes the problem into a priming sequence of fixed trajectories followed by navigation episodes, which allows to accelerate training by pre-computing priming representations.

**Sequence models** were early on implemented with recurrence in RNNs, LSTMs [26] and GRUs [16]. Their memory capacity scaling problem had been addressed by external neural memory [22, 55, 70], but was then eclipsed by transformers [59], which replaced recurrence altogether by attention. While attention over time still dominates multiple fields like NLP and CV, currently recurrence makes inroads again, either through state space models like S4 [24], Mamba [23] and LRU [45] inspired from control theory, or by combining it with attention: xLSTM [5], MooG [58] and Token Turing Machines [47] are prominent examples. We introduce a new transformer-based recurrent model and show that it scales favorably, can be trained for long sequences and generalizes to even longer ones.

**Relative pose estimation (RPE)** between two images was tackled for decades with pixel-level image matching techniques [25, 43, 49]; before learning-based approaches were proposed [29].

---

[1]This expresses complexity depending on the history size $N$ only.

Self-supervision was quickly introduced [36]. More recently, DUSt3R [60] regresses pointmaps, while MASt3R [35] additionally learns a descriptor inspired by image matching. Both leverage CroCo [62, 63] for pre-training. Similarly, MicKey [4] regresses pointmaps and supervises relative pose alone with differentiable RANSAC formulations [8, 11]. All these recent methods have led to impressive results for relative pose estimation, even under scenarios with little overlap between input images as in the MapFree-Relocalization benchmark [3]. Compared to the standard formulation, our "*Mem-RPE*" task requires estimating the pose between an image and the agent's memory.

## 3 The Mem-Nav and Mem-RPE tasks

**Mem-Nav** — We study navigation in photorealistic 3D environments, where an agent is given a goal image $\mathbf{g} \in \mathbb{R}^{3 \times H \times W}$ and is required to navigate from a starting location to the position shown in the goal. At each time step $t$ the agent observes a pair of sensor readings $\mathbf{o}_t = \{\mathbf{x}_t, \mathbf{u}_t\}$, where $\mathbf{x}_t \in \mathbb{R}^{3 \times H \times W}$ is an RGB image of size 112×112, and $\mathbf{u}_t \in \mathbb{R}^7$ is an odometry estimate in the form of a difference of agent poses between $t$ and $t{-}1$.

In contrast to classical navigation tasks in embodied AI, we model a continuous navigation setting by dividing each episode into two different parts: An initial *priming sequence* of length $P$, around 200 steps, in which the agent explores the scene and has access to observations $\{\mathbf{o}_t\}_{t=1...P}$ but *not yet* the future goal. During this initial sequence, the agent cannot chose its own actions and follows a predefined path. From step $P{+}1$ on, the agent receives the goal image $\mathbf{g}$ additionally to the observations $\{\mathbf{o}_t\}_{t=P+1...}$ and must navigate by predicting actions $\mathbf{a}_t$ at each step. The action space is the discrete set $\mathcal{A} =$\{move forward 0.25m, turn left $10°$, turn right $10°$, stop\}. An episode is considered successful if the agent calls the stop action within 1m of the goal position *and* within its 1000 steps budget. We use the Habitat simulator [48].

**Mem-RPE as an intermediate skill** — Navigating to previously seen positions efficiently requires the capacity to predict where they are, and we train our model for exactly this skill. We introduce the new sub-task of relative pose estimation between an agent position at time $t$, represented by a latent memory $\mathbf{m}_t$ maintained by the agent, and a query image $\mathbf{q}$, as $\mathbf{p} = \{\mathbf{t}, \mathbf{R}\}$, where $\mathbf{t} = \{d, \theta\}$ is the translation, *i.e.* distance and bearing angle from the agent to the position depicted in query image. $\mathbf{R}$ is the rotation matrix of the goal *towards* the agent, which is of limited relevance to a navigation task, but which we supervise to increase the learning signal during training.

## 4 Situation awareness with latent memory

We designed a new sequence model around the following goals targeting continuous robotics:

**(G1) Recurrence** — Classical auto-regressive models attending over a history of observations are required to revisit every single historical item for each step. Not only is this repetition wasteful in itself, it is further exacerbated by the quadratic computational complexity of transformers of $O(N^2)$ given $N$ observed items from the past. We target models maintaining a memory representation $\mathbf{m}_t$ updated at each step given the current observation $\mathbf{o}_t$ only, and queried directly, without again considering other historical information, leading to a complexity of $O(1)$ plus complexities arising from the distributed nature of the memory itself.
While transformers attending over time are currently the dominant models in embodied AI, and we do not claim to argue against their usage, in this paper we argue that the dominance might potentially be an overfit of the scientific process to the short episode lengths from existing benchmarks. While this is suitable for certain application which do not require long-term memory, like manipulation, we think that it does hold back research in areas where very long-term memory is necessary. Some applications in robotic navigation fall into this case, where it is advantageous to remember information seen minutes, hours, or even days ago.

**(G2) Memory capacity** — Holding actionable information about an entire observed scene requires scaling the memory size. Unfortunately, classical recurrent models like RNNs, LSTMs and GRUs are held back by the direct coupling of memory capacity and network capacity with a quadratic relationship determined by their update matrices. We address this by introducing a distributed hidden state $\mathbf{m}_t$ of $N$ embeddings of size $E$, and modeling the update function as a transformer. This ensures that the capacity of the network (transformer) can be chosen independently of the memory capacity by scaling $N$.

**(G3) Stability** — While we advocate for recurrent models for the reasons given above, they come with a shortcoming: training requires back-propagating gradients over memory update chains spanning over the sequence length. This is in contrast to transformers attending over time, where the sequence length is dealt with attention: while the number of attended items is as large as for recurrent models, the length of the gradient multiplication chain is not related to the context length, as items are not integrated *sequentially*, making these models more stable. We address this issue by combining research from classical recurrent models with an architecture from the transformer literature: we add gating functions to the memory update, allowing the model to take decisions (to "gate") on the speed of updates for each memory item.

In what follows, we introduce a new high-capacity recurrent model maintaining a latent representation of an observed scene, but as we will compare it to several recurrent baseline models from the literature in the experimental section, we start with quite general equations which fit all tested models. All considered models maintain some form of memory $\mathbf{m}_t$ over time steps $t$, and which are characterized by concrete implementations of the following functions:

$$
\begin{aligned}
\tilde{\mathbf{x}}_t &= Enc_{vis}(\mathbf{x}_t) && \text{// Encode visual input} \\
\tilde{\mathbf{u}}_t &= Enc_{odo}(\mathbf{u}_t) && \text{// Encode odometry input} \\
\mathbf{m}_t &= Update(\mathbf{m}_{t-1}, \tilde{\mathbf{x}}_t, \tilde{\mathbf{u}}_t) && \text{// Update memory} \\
\mathbf{y}_t &= Read(\mathbf{m}_t) && \text{// Read out memory} \\
\mathbf{p}_t &= Dec(\mathbf{y}_t, Enc_{goal}(\mathbf{g})), && \text{// Decode relative pose}
\end{aligned}
\tag{1}
$$

Starting with the commonalities, all models encode the visual input with a *Vision Transformer (ViT)* [20], $Enc_{vis}(\mathbf{x}_t)$, which we initialize with the weights of DINO-v2 [44] ViT-Small/14, and finetune the weights during training. Odometry inputs are encoded with an MLP, $Enc_{odo}(\mathbf{u}_t)$. All models update memory at each time step and also project memory $\mathbf{m}_t$ into a set of embeddings $\mathbf{y}_t = Read(\mathbf{m}_t)$ through a read-out mechanism. Both functions are tailored to each model, given further below. And lastly, we train all models in a supervised manner by predicting relative pose $\mathbf{p}_t$ with a decoder $Dec$, which is implemented as a transformer with cross-attention between encoded query image and read out memory, followed by self-attention,

$$
\begin{aligned}
\tilde{\mathbf{p}}_t &= CrossAttn(Q = Enc_{goal}(\mathbf{g}), K = \mathbf{y}_t, V = \mathbf{y}_t) \\
\mathbf{p}_t &= SelfAttn(\tilde{\mathbf{p}}_t).
\end{aligned}
\tag{2}
$$

For clarity we omitted residual connections and FF layers from the notation, pose is predicted from an additional CLS token added to the inputs.

## 4.1  Kinaema — memory in motion

We call our model "*Kinaema*", a neologism from *kinema* (motion) and *mnema* (memory). It maintains a set of $N$ embeddings $\mathbf{m}_t = \{\mathbf{m}_{t,n}\}$ of dimension $E$, and its update is recurrent (Goal **G1**) and implemented as transformer, ensuring that the memory size can be scaled by increasing $N$ without having to modify the network capacity (Goal **G2**). This is achieved by modeling the memory update $\mathbf{m}_t = Update(\mathbf{m}_{t-1}, \tilde{\mathbf{x}}_t, \tilde{\mathbf{u}}_t)$ as follows, also shown in Fig. 2. Each memory embedding $\mathbf{m}_{t,n}$ is summed with a learned positional embedding $\mathbf{e}_{t,n}$ combined with the encoded observations $(\tilde{\mathbf{x}}_t, \tilde{\mathbf{u}}_t)$ through concatenation, and then encoded, resulting in memory embeddings corrected (in a Kalman-like sense) by the observations:

$$
\mathbf{m}_{t,n}^{corr} = Linear([\mathbf{m}_{t-1,n} + \mathbf{e}_n, \tilde{\mathbf{x}}_t, \tilde{\mathbf{u}}_t]),
\tag{3}
$$

where $[.,.]$ denotes concatenation over the embedding dimension. The output size of the linear layer is $E$. While the last operation deals with each embedding independently, the following self-attention transformer contextualizes the embeddings with each other,

$$
\tilde{\mathbf{m}}_t = SelfAttn(\mathbf{m}_t^{corr}).
\tag{4}
$$

The resulting set of embeddings $\tilde{\mathbf{m}}_t$ corresponds to the update candidates, which are then subject to gating to increase training stability (Goal **G3**) and to model different speeds of dynamics in memory. The gating block operates on each embedding independently, giving

$$
\mathbf{m}_{t,n} = Gating(\mathbf{m}_{t-1,n}, \tilde{\mathbf{m}}_{t,n}).
\tag{5}
$$

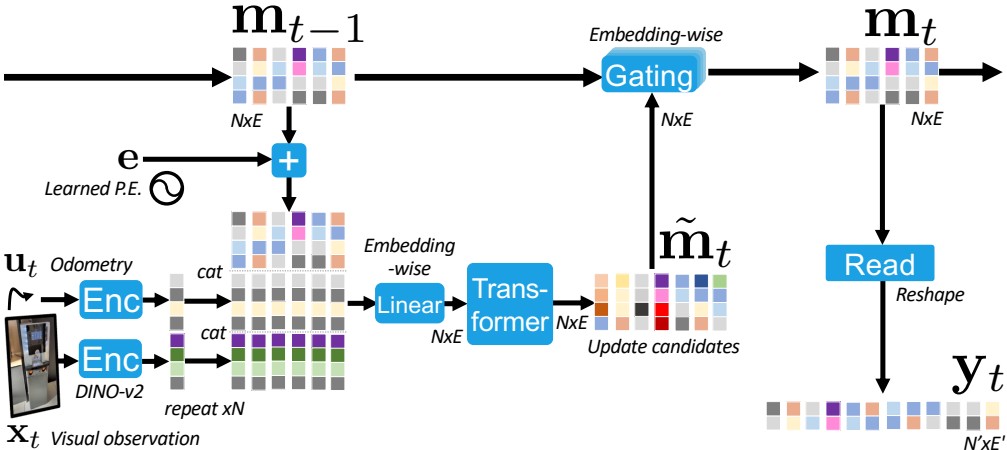

Figure 2: **Kinaema** is a recurrent sequence model maintaining distributed memory $\mathbf{m}_t$ in the form of $N$ embeddings of size $E$ each. Its previous state $\mathbf{m}_{t-1}$ is first contextualized with observations $\mathbf{o}_t = \{\mathbf{x}_t, \mathbf{u}_t\}$ then embedding-wise gated, resulting in new state $\mathbf{m}_t$.

Gating is inspired by GRUs by adding an update gate and a forget gate to the model [16], and we actually exploit this by implementing the gating block as a GRU cell with weights shared over the $N$ embeddings of $\mathbf{m}_t$. This comprises an essential difference between GRUs and Kinaema: in classical gated recurrent networks the gating mechanism is of the same complexity as the actual update function (the parameter matrices for the update and reset gate are of size identical to the matrix handling the update of the hidden state), which would be intractable for model as large as our model with a large distributed state. In *Kinaema*, the actual state update is handled through the transformer given in Eq. (4), whereas the gating is done by block weight-shared over memory embeddings. This choice allows to scale memory capacity easily without impacting the capacity of the gating block. In Sec. 5, we will show that both the transformer block in Eq. (4) and the gating block in Eq. (5) are essential for good performance.

The read-out block projects memory embeddings to a representation useful for the downstream decoders. For our model, this is implemented as a basic reshape, shaping the memory tensor from $N \times E$ to $N' \times E'$, $\mathbf{y}_t = Reshape_{N', E'}(\mathbf{m}_t)$. In our experiments, we will show, that it is interesting to have fewer memory of embeddings of higher embedding dimension, and to read them out into a larger number embeddings of lower dimension for decoding.

**Training** — We train on sequences of randomized lengths $T$ between 50 and 100 time steps, and by taking the memory $\mathbf{m}_T$ at the last time step, predicting relative pose for $2T$ query images of two different types: (i) the $T$ observed images $\{\mathbf{x}_t\}_{1...T}$, and (ii) $T$ *alternative* images $\{\mathbf{x}_t^{alt}\}_{1...T}$, which have *not* been observed but lie in the observed region of the scene. We generate them with the same simulator by slightly disturbing the pose of the observation of the corresponding time step $t$. These frames prevent the model to learn a simple lookup table. We train with supervision, $\mathcal{L}_{RPE} = \sum_i \left[ |\mathbf{t}_i - \mathbf{t}_i^*| + |\mathbf{R}_i - \mathbf{R}_i^*| \right]$, where $(\mathbf{t}_i, \mathbf{R}_i)$ and $(\mathbf{t}_i^*, \mathbf{R}_i^*)$ are predicted and GT pose for training image $i$, respectively. We add an auxiliary masked image modeling loss, which reconstructs the same query images after they have been masked. We add a second decoder head querying memory with the same cross-attention mechanisms as the RPE decoder in Eq. (2) — see Appendix B.

### 4.2 Integration into the downstream navigation agent

We address the Mem-Nav task introduced in Sec. 3 by augmenting the *DEBiT* agent from [9], which currently achieves state-of-the-art performance on the *ImageNav* and *Instance-ImageNav* tasks:

$$
\begin{aligned}
\mathbf{p}_t^{obs} &= BinEnc(\mathbf{x}_t, \mathbf{g}) && \text{// Binocular encoder - get goal direction} \\
\mathbf{h}_t &= GRU(\mathbf{h}_{t-1}, \mathbf{p}_t^{obs}, RN(\mathbf{x}_t), MLP(\mathbf{a}_{t-1})) && \text{// Recurrent memory update} \\
p(\mathbf{a}_t) &= \pi(\mathbf{h}_t). && \text{// Linear policy}
\end{aligned}
\tag{6}
$$

This agent maintains a recurrent GRU memory $\mathbf{h}_t$ fed with visual observations $\mathbf{x}_t$ encoded by a ResNet-18. More importantly, it compares each visual observation $\mathbf{x}_t$ with the goal image $\mathbf{g}$ using a

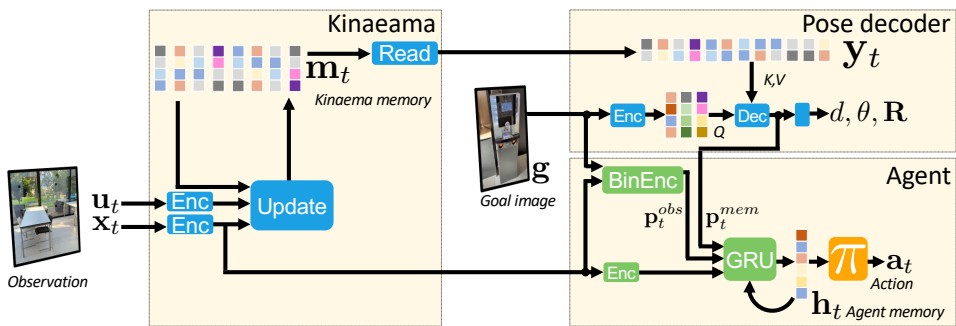

Figure 3: **Integration into the downstream RL-trained agent**: The RPE decoder used for pre-training is kept for the downstream task, searching for the goal image in the embeddings $\mathbf{y}_t$ which are read out from memory $\mathbf{m}_t$, while a binocular encoder $BinEnc$ from *DEBiT* [9] is used to compare the goal to the current observation. **Two types of memory are updated and queried**: (i) the agent maintains its own additional recurrent memory $\mathbf{h}_t$, and (ii) Kinaema-memory $\mathbf{m}_t$.

binocular transformer $BinEnc$, pre-trained for relative pose and visibility estimation between pairs of images, to extract information on the goal direction.

The *DEBiT* agent is capable of efficiently detecting goals when they are visible, but does not handle previously seen goals, in particular when they have been observed *before* an episode start. This is where our proposed new sequence model comes in — it is run in parallel, receives the same visual observations $\mathbf{x}_t$ as the main agent, and contributes with goal direction estimates.

As illustrated in Fig. 3, the augmented agent is given as

$$
\begin{aligned}
\mathbf{p}_t^{obs} &= BinEnc(\mathbf{x}_t, \mathbf{g}), && \text{// Binocular encoder - get goal direc} \\
\mathbf{m}_t &= \ldots \text{using Eqs.(1)}, && \text{// Kinaema memory update} \\
\mathbf{p}_t^{mem} &= \ldots \text{using Eqs.(1)}, && \text{// Kinaema - get goal direction} \\
\mathbf{h}_t &= GRU(\mathbf{h}_{t-1}, \mathbf{p}_t^{obs}, \mathbf{p}_t^{mem}, RN(\mathbf{x}_t), MLP(\mathbf{a}_{t-1})), && \text{// Recurrent memory update} \\
p(\mathbf{a}_t) &= \pi(\mathbf{h}_t). && \text{// Linear policy}
\end{aligned}
\tag{7}
$$

where $\mathbf{p}_t^{mem}$ is given by the relative pose decoder of the *Kinaema* model, denoted as $\mathbf{p}_t$ in Eq. (1). Inspired by [9], for both pose estimates, $\mathbf{p}_t^{obs}$ and $\mathbf{p}_t^{mem}$, we provide the latent encoding of the pose taken from the penultimate layers of the respective networks, and not the decoded pose values themselves.

**Training** — we train the parameters of the policy $\pi$, the recurrent network $GRU$ and the monocular ResNet encoder $RN$ jointly with PPO [50] for 300M steps, with a reward definition in the lines of the one proposed by [15] for *PointGoal* and re-used by [9] for *ImageGoal*, $r_t = \text{K} \cdot \mathbf{1}_{\text{success}} - \Delta_t^{\text{Geo}} - \lambda$, where $K{=}10$, $\Delta_t^{\text{Geo}}$ is the increase in geodesic distance to the goal, and slack cost $\lambda{=}0.01$ encourages efficiency.

We initialize the agent from publicly available trained DEBiT-B model provided by [9], and train the remaining parameters from scratch. Since the size of the GRU was increased by the additional inputs, of the extended agent, this required a block-wise initialization of the weight matrix which project GRU input to latent memory space:

$$
\mathbf{W} = \begin{bmatrix}
\mathbf{W}_{Kinaema \to h} &=& \mathbf{0} \\
\mathbf{W}_{monoc \to h} &=& \mathbf{W}_{monoc \to h} \text{ pre-trained from [9]} \\
\mathbf{W}_{binoc \to h} &=& \mathbf{W}_{binoc \to h} \text{ pre-trained from [9]} \\
\mathbf{W}_{act \to h} &=& \mathbf{W}_{act \to h} \text{ pre-trained from [9]}
\end{bmatrix}
$$

Just after initialization, the extended agent gives the exact same outputs as DEBiT-B [9]. We then re-initialize the linear policy layer with default uniform distribution.

## 5 Experimental results

**Experimental setup** — We trained the models on the HM3D [46] and Gibson [66] datasets and created a sound set of splits allowing clean separation of pre-training, training and evalua-

| Model | Mem size | Obs hist | Seq len 200 | | | Seq len 800 | | |
|---|---|---|---|---|---|---|---|---|
| | | | $1m$ $10^o$ | $1m$ $90^o$ | $2m$ $90^o$ | $1m$ $10^o$ | $1m$ $90^o$ | $2m$ $90^o$ |
| *Trunc.Hist.* | 41.6k | ✓ | 2 | 11 | 28 | 1 | 6 | 16 |
| **MooG** [58] | 524.3k | ✗ | 0 | 5 | 14 | 0 | 3 | 9 |
| **LRU** [43] | 3.1k | ✗ | 4 | 18 | 34 | 2 | 9 | 20 |
| **EMA** [21] | 153.6k | ✗ | 6 | 18 | 34 | 3 | 11 | 24 |
| **xLSTM** [5] | 2,359.3k | ✗ | 8 | 23 | 47 | 5 | 13 | 29 |
| **GRU** [16] | 3.1k | ✗ | 12 | 32 | 56 | 4 | 14 | 31 |
| *Kinaema* | 61.4k | ✗ | 21 | 41 | 63 | 10 | 21 | 37 |

Table 1: **Comparisons of models** on Mem-RPE: *Kinaema* has $N{=}20$ memory embeddings; EMA uses a trainable $\lambda$ ( **RPE-test** ).

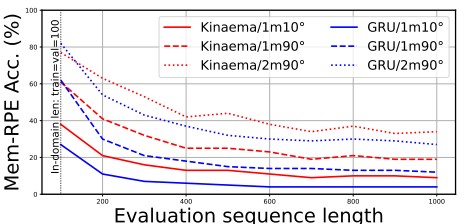

Figure 4: **Generalization to longer sequences, Mem-RPE**: GRU and *Kinaema*, trained for T=100, evaluated on $T{=}100...1000$ ( **RPE-test** ).

tion: **RPE-train** was used for Mem-RPE training and consisted of sequences sampled from the HM3D/train scenes. **RPE-val** contains sequences generated from Gibson/train scenes and was used for checkpoint selection and ablations. **RPE-test** contains sequences generated from HM3D/val scenes and was used for final evaluation and model comparisons. **NAV-train** was used for navigation training on new episodes generated from the HM3D/train scenes: starting poses match pre-generated offline priming sequences. A similar arrangement was done for **NAV-test** , based on HM3D/val scenes. All tables have color-coded backgrounds indicating the splits. More details in Appendix A.

**Metrics / Mem-RPE** — All models have been trained with randomized sequence lengths sampled between $T{=}50$ and $T{=}100$. We systematically evaluate all models in significant out-of-distribution settings, generalizing to two different validation sequence lengths of $T{=}200$ and $T{=}800$, respectively. We provide accuracy of correctly recognized poses with three tolerance margins: less than $1m$ of translation and $10°$ of rotation errors, $< 1m$ and $90°$, and $< 2m$ and $90°$. We put more emphasis on low translation errors, almost disregarding goal rotation: this is goal rotation towards the agent, irrelevant for navigation. Rotation *towards* the goal, a.k.a. "*bearing*", is part of the translation error.

**Metrics / Mem-Nav** — Navigation performance is evaluated by success rate (SR), *i.e.*, fraction of episodes terminated within a distance of $<$1m to the goal by the agent calling the stop action, and SPL [2], *i.e.*, SR weighted by the optimality of the path, $SPL = \frac{1}{N} \sum_{i=1}^{N} S_i \frac{\ell_i^*}{\max(\ell_i, \ell_i^*)}$, where $S_i$ be a binary success indicator in episode $i$, $\ell_i$ is the agent path length and $\ell_i^*$ the shortest path length.

**Baseline models** — We implemented the following baseline sequence models, which were adapted to the task by adding a read-out mechanism and the same RPE-decoder described in Sec. 4, Eq. (2). More details on them is given in Appendix D.

**GRUs** [16] model memory $\mathbf{m}_t$ as a single vector and were implemented with the standard PyTorch implementation, $Update \triangleq \mathbf{m}_t = \mathbf{W}\mathbf{m}_{t-1} + \mathbf{U}\tilde{\mathbf{x}}_t$, where we omitted gating equations from the notation. We explored multiple numbers of layers and we made the memory read-out function non-linear with (non-shared) MLPs, $Read(\mathbf{m}_t) \triangleq \{MLP_{\theta_i}(\mathbf{m}_t)\}_i$.

**EMA** [21] models memory update as an exponential average, $Update \triangleq \mathbf{m}_t = \lambda\mathbf{m}_{t-1} + \mathbf{U}\mathbf{x}_t$, with the readout being a simple reshape. They are simple but provide interesting guarantees; more importantly, lacking any learned dynamics, they allow to evaluate the impact of significantly increasing memory capacity without having to deal with side-effects on stability.

**xLSTMs** [5] maintain a matrix shaped cell state updated with the covariance update rule [51] resorting to cross-products of K and V projections. We took the official code [2] and projected the models hidden memory to a set of embeddings with the parallel MLP chain given above. This uses xLSTMs "as is", *i.e.* as a standard sequence model. Potential further integration could be done by opening the black box and adapting the internal querying mechanism to the task.

**MooG** [58] is probably the model closest to us, as it is recurrent with a transformer update. However, it has key differences: there is no gating block, updates separate a prediction and a correction step, inputs are dealt with patch-wise and cross-attended to 1024 memory embeddings of size 512, learned embeddings are replaced with memory initialization. We re-implemented it and adapted by giving it the same inputs (including $\mathbf{u}_t$) as the other models. This re-implementation has the exact architecture and hyper-parameters as described in the paper, but for comparability does not use their specific loss, which separates a prediction and a correction step.

---

[2] https://github.com/NX-AI/xlstm

**Trunc.Hist.** is a simple baseline which forwards the last $T_{\text{trunc}}$ observations embeddings directly to the decoder. It is not recurrent and has a limited context length.

**DEBiT** [9] is a natural baseline for the navigation task. It was described in Sec. 4.2 and currently holds SoTA performance on *ImageNav* and *Instance-ImageNav*. We used the official code.[3]

**Mem-RPE performance** — Tab. 1 compares *Kinaema* with the baselines on the *Mem-RPE* task. **EMA** has a massive advantage in terms of memory size, which we configured to $|\mathbf{m}_t|=400*384=153.6k$, but this could not compensate its very simple dynamics modeled as exponential decay. As its memory content cannot be re-arranged by the *Update* function, the burden of organizing it lies with the input encoder placing inputs correctly into the memory values. Compared to the original [21], we made $\lambda$ a trainable vector of size $153.6k$, which boosted performance (see Appendix E for the performance of [21]). **GRU** clearly outperforms EMA due to its more expressive handling of process dynamics. Given its small memory capacity, it was crucial to encourage the model to compress memory, which we achieved by making *Read* non-linear. This performed considerable better than the linear variants (see Appendix E). The more recent **xLSTM** performed less well than a GRU, but was used as a plug-n-play sequence model. We conjecture that performance could be optimized further by opening the black box and making the internal query mechanism connect to the goal image more directly. **MooG** was reported to be trained for $T=8$ steps only in [58] but we trained it the same lengths of $T=100$ as the other models in our experiments. It performed very poorly, which we link to the patch-wise handling of visual inputs, which seems to overwhelm the recurrent transformer. Out attempts to switch *Kinaema* to a similar handling of memory and attention failed similarly. *Kinaema* has been configured such that each embedding is of the same size as the GRU memory, and the model can leverage its larger memory, making use of its multiple embeddings. It provided the best performance, in particular when needing the generalize to longer sequences. We link this to the combination of large memory size, expressive transformer update, and stable training provided by the gating block. **Trunc.Hist.** was trained with $T_{\text{trunc}}=100$, and evaluated on longer sequences by truncation. It does not generalize well, and delegates all the work to the decoder, which lacks capacity.

All models were trained on a max seq. length of $T=100$, we see that performances drop when they are evaluated on significantly larger lengths, $T=800$. Fig. 4 compares *Kinaema* with GRU on lengths 100...1000, where we see a sharp initial drop increasing $T$ from the in-domain value of $T=100$, followed by a more shallow further decrease. Comparisons of all models are given in Appendix E.

While the EMA and GRU models were quite stable during training, *Kinaema* followed a bi-modal distribution over seeds: seeds either gave excellent or mediocre performance. Results in Tab. 1 were given on **RPE-test** with seeds selected on **RPE-val** .

**Sensitivity study: memory capacity** — In Tab. 2a, we studied the impact of the number $N$ of embeddings in $\mathbf{m}_t$ of size $3072 = \sim 3k$. The model scales well until roughly $N=20$ embeddings are reached. We conjecture that this is due to training with sequences of length $T=100$ and longer training could lead to bigger choice for optimal memory usage. In Tab. 2b we studied the compromise between the number of embeddings and their dimensions for a given fixed memory size. Fewer and bigger embeddings seem to work better, which we tentatively explain by the factorization of the gating block of the model: each scalar gate value is determined as a function of the values of the same embedding. Increasing the embedding dim increases expressivity.

| Num emb | Emb dim | Mem size | Seq len 200 $1m$ $10^o$ | $1m$ $90^o$ | $2m$ $90^o$ | Seq len 800 $1m$ $10^o$ | $1m$ $90^o$ | $2m$ $90^o$ |
|---|---|---|---|---|---|---|---|---|
| 1 | 3.1k | 3.1k | 7 | 26 | 53 | 3 | 12 | 34 |
| 5 | 3.1k | 15.4k | 14 | 34 | 72 | 7 | 21 | 44 |
| 10 | 3.1k | 30.7k | 14 | 36 | 62 | 9 | 23 | 45 |
| 20 | 3.1k | 61.4k | 24 | 52 | 77 | 13 | 28 | 47 |
| 30 | 3.1k | 92.2k | 24 | 51 | 73 | 9 | 22 | 40 |
| 50 | 3.1k | 153.6k | 23 | 47 | 68 | 1 | 7 | 22 |

(a)

| Num emb | Emb dim | Mem size | Seq len 200 $1m$ $10^o$ | $1m$ $90^o$ | $2m$ $90^o$ | Seq len 800 $1m$ $10^o$ | $1m$ $90^o$ | $2m$ $90^o$ |
|---|---|---|---|---|---|---|---|---|
| 160 | 384 | 61.4k | 4 | 19 | 44 | 2 | 13 | 33 |
| 80 | 768 | 61.4k | 2 | 12 | 31 | 1 | 7 | 20 |
| 40 | 1.5k | 61.4k | 6 | 26 | 52 | 3 | 14 | 33 |
| 20 | 3.1k | 61.4k | 24 | 52 | 77 | 13 | 28 | 47 |

(b)

Table 2: *Kinaema*: **varying memory structure**, Mem-RPE: (a) keeping memory embedding size constant; (b) keeping total memory size constant ( **RPE-val** ).

**Ablation studies** — Tab. 3 ablates the two main blocks of *Kinaema*'s *Update* block: both the transformer and the gating block are necessary for good performance. In Tab. 4 we ablate training choices. Randomizing sequence length $T$ during training is a key design choice. We found that training with constant lengths hindered generalization to longer sequences. We conjecture that it leads to models confusing the notion of "state" (in a control theory sense) with "layer of abstraction",

---

[3] https://github.com/naver/debit

| Transf. block | Gating block | Seq len 200 | | | Seq len 800 | | |
|---|---|---|---|---|---|---|---|
| | | $1m$ $10^o$ | $1m$ $90^o$ | $2m$ $90^o$ | $1m$ $10^o$ | $1m$ $90^o$ | $2m$ $90^o$ |
| ✗ | ✓ | 11 | 33 | 55 | 3 | 9 | 19 |
| ✓ | ✗ | 11 | 37 | 62 | 4 | 15 | 31 |
| ✓ | ✓ | 24 | 52 | 77 | 13 | 28 | 47 |

Table 3: *Kinaema* **ablations of update blocks**, impact on Mem-RPE ( **RPE-val** ).

| Randomize seq len | Masked img modeling | Seq len 200 | | | Seq len 800 | | |
|---|---|---|---|---|---|---|---|
| | | $1m$ $10^o$ | $1m$ $90^o$ | $2m$ $90^o$ | $1m$ $10^o$ | $1m$ $90^o$ | $2m$ $90^o$ |
| ✗ | ✓ | 8 | 30 | 58 | 4 | 15 | 36 |
| ✓ | ✗ | 28 | 51 | 71 | 4 | 14 | 34 |
| ✓ | ✓ | 24 | 52 | 77 | 13 | 28 | 47 |

Table 4: *Kinaema* **ablations of different losses**, impact on Mem-RPE ( **RPE-val** ).

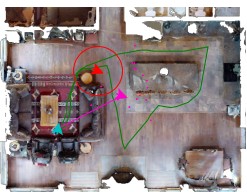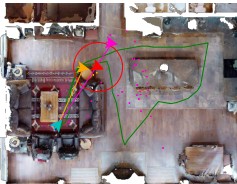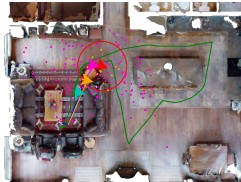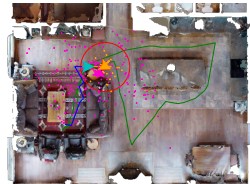

Figure 5: **RPE during navigation**: exploiting the information from the priming sequence (green), *Kinaema* can predict (pink) the rel. pose of the goal (red), while the binocular module of [9] only starts providing reliable predictions (orange) when the agent is positioned (cyan) in view of the goal.

*i.e.* using recurrent updates not only to push representations forward in time, but also to make changes in abstraction levels as a neural network would do between layers. The removal of masked image modeling particularly impacts OOD behavior, generalization to longer sequences.

**Navigation performance** — Three variants of the DEBiT agent [9], with different ways to integrate memories, are fine-tuned for 100M steps on **NAV-train** , and compared against the original agent on **NAV-test** in Tab. 5: (with *Kinaema*) concatenates the output of the *Kinaema* for Mem-RPE fed with primer and updated with observations to the input of the agent GRU, as described in Sec. 4.2; (with GRU memory) same, but using a GRU; (with $h_0$-injection) is a baseline using the hidden state $\mathbf{h}_t$ of the original agent to encode the priming sequence with no goal, teacher forced, and injects it into $h_0$ at episode start with linear adaptation.

| SPL (%) | DIST SPLIT → | EASY | (2-5m) | MEDIUM | (5-10m) | HARD | (10-22m) |
| | SEEN SPLIT → | SEEN | UNSEEN | SEEN | UNSEEN | SEEN | UNSEEN |
| ↓ MODEL | PRIMER ↓ | (#=566) | (#=116) | (#=520) | (#=319) | (#=64) | (#=290) |
|---|---|---|---|---|---|---|---|
| • DEBiT (zeroshot) | ✗ | 41 | 41 | 40 | 42 | 47 | 41 |
| • DEBiT (finetune) | ✗ | 45 | 40 | 42 | 43 | 48 | 44 |
| • w/. inject $h_0$ | ✗ | 37 | 32 | 35 | 37 | 42 | 38 |
| | ✓ | 41 (+4) | 40 (+8) | 36 (+1) | 38 (+1) | 41 (-1) | 40 (+2) |
| • w/. GRU memory | ✗ | 43 | 42 | 42 | 43 | 50 | 45 |
| | ✓ | 45 (+2) | **48** (+6) | 43 (+1) | 46 (+3) | 50 (+0) | 45 (+0) |
| • w/. Kinaema | ✗ | 48 | 47 | 46 | 45 | 44 | 48 |
| | ✓ | **50** (+2) | 46 (-1) | **48** (+2) | **49** (+4) | **54** (+10) | **49** (+1) |

Table 5: **Downstream navigation performance** on *Mem-Nav* ( **NAV-test** ): GRU and *Kinaema* are integrated into the agent as in Section 4.2. The baseline "$h_0$-injection" uses the non-augmented DEBiT agent over the priming sequence (teacher-forced) and episode (policy taking decisions) with the same state $\mathbf{h}_t$. See Fig. 10 in supp. mat. for the corresponding plot.

As shown in Tab. 5, naively adapting hidden state of the original agent *($h_0$ injection)* to represent primer information does not work. Using dedicated memory encoders significantly improves navigation performance, indicating that the agent could exploit the information to localize the goal and optimize its path. Fig. 10 shows navigation performance, measured by SPL, for different episode difficulties, measured by geodesic distance between start and goal. *Kinaema* offers a significant advantage compared to other models. While easier episodes can be solved by any model, the additional memory and RPE decoders are helpful when dealing with longer sequences.

In particular, we can observe the advantage of having obtained an initial exploration of the scene through the priming sequence (marked in the 2nd column) and stored in Kinaema memory.

Fig. 5 visualizes Mem-RPE predictions during navigation episodes, indicating that *Kinaema* can successfully predict the goal position even when it is not in view.

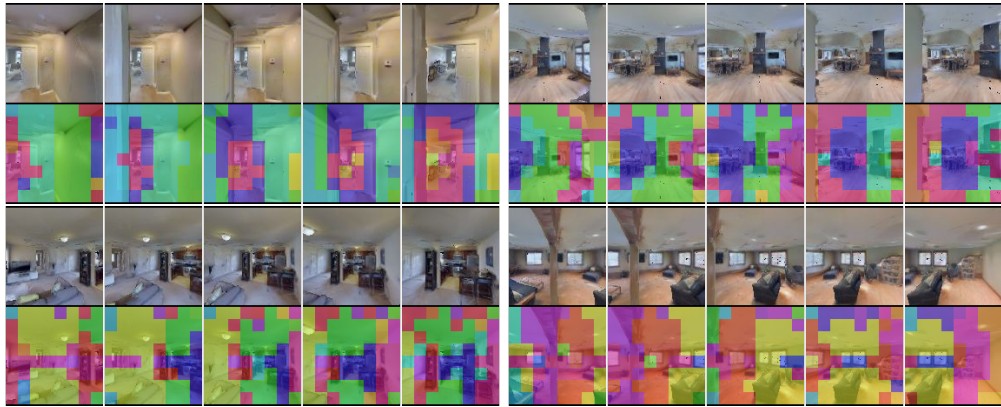

Figure 6: **Cross-attention over memory**: each image patch is colored by the memory embedding receiving the highest attention, revealing stable region–memory correspondences over time.

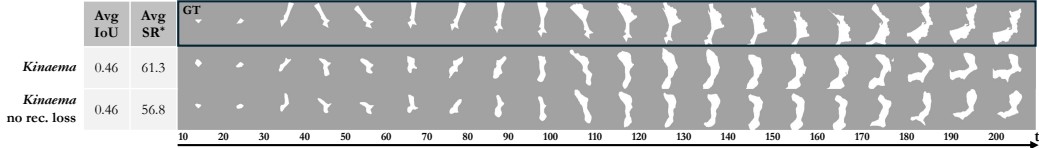

Figure 7: **Occupancy probing** of a sequence of length $T=200$ from $\mathbf{m}_t$. SR*: success rate on 10 navigation episodes defined on each GT map, but executed on the probed map.

**Visualizing cross-attention between memory and scene patches** — As mentioned in Eq. (2), the RPE decoder of our *Kinaema* model contains a cross-attention layer that operates between the patch tokens of the query images and 160 embeddings (20 memory embeddings $\mathbf{m}_t$ of size 3072 reshaped into 160 embeddings $\mathbf{y}_t$ of size 384). In Fig. 6, we visualize attention probabilities from this layer on four random selected Gibson episodes, each comprising 200 steps. There, each memory token is given a fixed color. Then for each query image, we associate its patch tokens to the most attended memory token, and overlay the corresponding color on the image. We see that patch tokens from spatially coherent regions tend to map to the same memory token, producing a segmentation-like effect. This correspondence is stable over time: as the agent moves, the set of patch tokens associated with a given memory token "moves" consistently with the viewpoint.

**Probing occupancy** — In Fig. 7 we show predictions of a probing model outputting occupancy BEVs from the frozen memory $\mathbf{m}_t$ of *Kinaema*. The probe is able to infer scene structures from Mem-RPE pre-training. While map reconstruction accuracy is the same with and without reconstruction loss (average IoU on the validation set), removing the loss degrades navigation performance in this setting: the avg. SR* using probed maps goes from 61.3% to 56.8%. More details are given in Appendix F.

**Limitations** — (i) *Kinaema* has been trained for relative pose estimation, but does not provide an estimate on whether a goal image has been seen in the past. (ii) Training has been limited to $T=100$ steps, an increase would likely improve performance. (iii) Dealing with visual inputs as a single embedding instead of a patch-wise representation enabled its high performance, compared to models like MooG [58], but it might limit adding further improvements in the future.

## 6 Conclusion

We have proposed a new recurrent sequence model maintaining a distributed memory updated with a transformer. Compared to transformers attending over the observation history, it is computationally efficient with updates and reads of $O(1)$. We trained the model for a new skill "*Mem-RPE*", relative pose estimation between a goal image and agent memory, and integrated it into a new continuous navigation downstream task, "*Mem-Nav*". We show that the model can spatially situate previously seen spaces and leverage this capability to navigate efficiently in a continuous operation. The model widely outperforms other recurrent baselines including recent work using transformer updates.

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

# Contents

## A    Data splits and generation

### A.1  RPE data

Pre-training data for relative pose estimation (RPE) is generated on scenes from the train split of the HM3D dataset. Using a navigating agent configuration close to the default one of habitat-sim, we collect a fixed number of frames per scene by chaining random goal pursuits. We split data per connected components of the navigation mesh, such that any sample of a subset can be reached from any other. We take care of balancing the number of samples with respect to the component area relative to the total navigable area of the scene. Compared to the default agent configuration, we use a smaller image resolution ($112 \times 112$) and an action space enabling finer motion, which is *deliberately different* from the action space of the agent used for the validation and testing data, for both Mem-RPE and Mem-Nav. This allows to test our model in out-of-distribution situations:

**Configuration 1: RPE-training** : $\mathcal{A} = \{10\text{cm forward}, 5° \text{ turns}\}$. This choice corresponds to the raw data generated. Additionally, during training frames are not sampled consecutive, but randomly in steps between 1 and 8.

**Configuration 2: RPE-validation+test, Mem-Nav training+test** :
$\mathcal{A} = \{25\text{cm forward}, 10° \text{ turns}\}$.

This generates significant out-of-distribution behavior between Mem-RPE training and Mem-RPE testing. Note that a forward testing step of 25cm *cannot* be expressed as multiple steps of forward training steps of 10cm.

Additionally, one alternative frame is generated at each step by randomizing most of the sensor specification. We provide details of the parameter distributions in Tab. 6. Algorithm 1 provides more details on the generation procedure. We use a custom storage format based on a compressed NumPy archive supporting both dense and sparse values (*e.g.* for the pursued goal position), and large folders of images compressed in JPEG. Validation and test data are generated on different scenes with the same procedure. We use the train split of the Gibson dataset for validation, from which we select hyper-parameters and models without interfering with the tests used to report metrics. The final metrics are reported on the HM3D val split (the test split is not publicly available). However, as detailed in Tab. 6, agent configuration uses the standard action space instead (25cm forward, $10°$ turns), to match the one of the downstream navigation task.

---

**Input:** scene dataset $\mathcal{S}$, #frames per scene
**Output:** Offline dataset of trajectories with main and alt cam
**foreach** *scene in $\mathcal{S}$* **do**
   split scene into connected navigable islands;
   select subset $\mathcal{I}$ of islands to get $80\%$ of scene coverage;
   **foreach** *island in $\mathcal{I}$* **do**
      #frames $\leftarrow$ area(island) / total covered area $\times$ #frames per scene;
      agent state $\leftarrow$ random navigable position, random orientation;
      **repeat**
         select random navigable goal with geodesic dist. constraints;
         **while** *goal not reached* **do**
            agent state $\leftarrow$ execute greedy action towards goal;
            alternative camera $\leftarrow$ random configuration;
            render main and alternative frames;
            store(frames, agent position and orientation, alt camera);
         **end**
      **until** *#frames have been generated*;
   **end**
**end**

**Algorithm 1:** RPE data generation procedure.

## A.2 MemNav data

For the downstream navigation task, we only use the scenes in the publicly available train and val splits of the HM3D dataset. We use the default agent configuration of habitat-sim except for the frame resolution which we reduce to $112 \times 112$ to avoid unnecessary rendering and inter-process transfer for frames ending up resized or max-pooled in the agent model. We first sample a few exploration cycles per connected component of the navigable mesh in each scene, to constitute limited offline information used as memory primers for the agent, and for which we can pre-compute internal representation. Start poses are then sampled from these cycles with many different goal positions to make navigation episodes. Algorithm 2 provides details of the generation procedure, and Algorithm 7 lists important generation parameters.

## B   More details on training losses

As mentioned in Sec. 4.1 of the main paper, we train all models by jointly optimizing the relative pose estimation (RPE) and masked image modeling (MIM) losses over sequences of length $T = 100$.

Mean-squared error is used for the MIM loss:

$$\mathcal{L}_{MIM} = \frac{1}{N \times P} \sum_{i=1}^{N} \sum_{p \in I_i} \Big[ |\mathbf{p}_i - \mathbf{p}_i^*|_2^2 \Big], \tag{8}$$

where $\mathbf{p}_i$ and $\mathbf{p}_i^*$ are predicted and GT pixel values for the patch $p$ of the training image $i$, and $N$ and $P$ are the number of training images and patches in each image, respectively.

| | | | | |
|---|---|---|---|---|
| **Common** | cameras | main | resolution | $112 \times 112$ |
| | | | field of view | $90°$ |
| | | | position | 125cm above ground |
| | | | orientation | straight |
| | | alt | resolution | $112 \times 112$ |
| | | | field of view | $\sim \mathcal{U}[60°, 120°]$ |
| | | | aspect ratio | $\sim \mathcal{U}\{1, 4/3, 16/9, 16/10\}$ $\times$ (portrait or landscape) |
| | | | position | 125cm above ground $+\Delta \sim \mathcal{U}[\pm 50cm]$ in all dir. |
| | | | orientation | pan $\phi \sim \mathcal{U}[\pm 50°]$ tilt $\psi \sim \mathcal{U}[\pm 30°]$ roll $\theta \sim \mathcal{U}[\pm 5°]$ |
| **RPE-train** | actions | forward | 10cm | |
| | | turn L/R | $5°$ | |
| | scenes | set | HM3D/`train` | |
| | | # | 800 | |
| | frames | # | per scene | 100k |
| | | | total | 80M |
| | sampling | rand. intervals | skip | $\sim \mathcal{U}[0, 8]$ frames |
| **RPE-val** | actions | forward | 25cm | |
| | | turn L/R | $10°$ | |
| | scenes | set | gibson/`train` | |
| | | # | 72 | |
| | frames | # | per scene | 1k |
| | | | total | 72k |
| | sampling | contiguous seq. | | |
| **RPE-test** | actions | forward | 25cm | |
| | | turn L/R | $10°$ | |
| | scenes | set | HM3D/`val` | |
| | | # | 100 | |
| | frames | # | per scene | 1k |
| | | | total | 100k |
| | sampling | contiguous seq. | | |

Table 6: **Parameters for RPE data generation.**

The RPE loss is given in Sec. 4.1 of the main paper, and repeated here for completeness:

$$\mathcal{L}_{RPE} = \frac{1}{N} \sum_{i=1}^{N} \Big[ |\mathbf{t}_i - \mathbf{t}_i^*| + |\mathbf{R}_i - \mathbf{R}_i^*| \Big], \tag{9}$$

where $(\mathbf{t}_i, \mathbf{R}_i)$ and $(\mathbf{t}_i^*, \mathbf{R}_i^*)$ are predicted and GT pose for the training image $i$, respectively.

See Appendices C.2 and C.3 on how we make RPE and MIM predictions, respectively.

**Generalization to longer sequences** — We apply two types of "drop-out" to improve generalization between train and test, and generalization to longer sequences:

1. random sequence sub-sampling, where a sequence length is randomly determined at each training iteration to lie between a minimum value ($T_{\text{sub}}$) and $T$.

2. as already mentioned in section A.1 on data generation, we do not sample consecutive steps from the generated training data, but rather use random interval sampling of size 8, where the interval between two consecutive steps can be up to 8. This encourages robustness to changes in sampling rate between train and test.

**Training hyper-parameters** commonly used for all models are shared in Tab. 8.

**Input:** scene dataset $\mathcal{S}$, #cycles per island, #frames per cycle, #episodes per scene
**Output:** "navigate from memory" dataset of episodes with primers
**foreach** *scene in $\mathcal{S}$* **do**
> split scene into connected navigable islands;
> select subset $\mathcal{I}$ of islands to get $80\%$ of scene coverage;
> **foreach** *island in $\mathcal{I}$* **do**
>> **for** *#cycles per island* **do**
>>> **while** *approximate cycle length $<$ #frames per cycle* **do**
>>>> add random waypoint to cycle;
>>>
>>> **end**
>>> solve TSP on waypoints;
>>> agent state $\leftarrow$ arbitrary waypoint in cycle;
>>> **foreach** *waypoint in cycle* **do**
>>>> **while** *waypoint not reached* **do**
>>>>> agent state $\leftarrow$ execute greedy action towards waypoint;
>>>>> render main camera frame;
>>>>> ray-trace fog of war on occupancy grid;
>>>>> store(frame, fog, agent position and orientation);
>>>>
>>>> **end**
>>>
>>> **end**
>>> #episodes $\leftarrow$ area(island) / total covered area / #cycles $\times$ #frames per scene;
>>> **for** *#episodes* **do**
>>>> start (pos, ornt) $\leftarrow$ random pose along cycle;
>>>> goal (pos) $\leftarrow$ random navigable point;
>>>> memory primer $\leftarrow$ sub-sample cycle (frames,poses) to fixed #frames per cycle;
>>>> /* such as memories end at start pose                    */
>>>> geodesic distance $\leftarrow$ shortest path length from start to goal;
>>>> seen indicator $\leftarrow$ approx. goal visibility from cycle (fog or neighbor alignment);
>>>> store(episode);
>>>
>>> **end**
>>
>> **end**
>
> **end**

**end**

**Algorithm 2:** MemNav data generation procedure.

## C  Kinaema: architecture details

### C.1  Main memory model

**Inputs**: images are of resolution $112 \times 112$.

**Visual encoders** ($Enc_{vis}$ and $Enc_{goal}$): they are implemented as ViT-Small with a patch size of 14 and 0 registers [17], and they are initialized from the pretrained weights of DINO-v2 [44].

**Odometry encoders** ($\tilde{u}_t$): they are implemented as linear layers with an output size of 64.

**Update/correction block**: equation (3) is Linear function.

**Update/transformer block**: the transformer block has 3 layers, 24 heads, and an MLP-factor of 4. This was optimized over the validation set **RPE-val**, exploring 0,1,2,3,4 layers and various numbers of heads.

**Update/gating block**: the gating block is a GRU in the standard PyTorch implementation with 3 layers. We explored 0,1,2,3 and 4 layers, optimized over **RPE-val**.

### C.2  RPE modules

All models have a Transformer-based decoder to provide estimates for relative pose, but there are slight differences, as we optimized this for the baselines. In order words,

| common | cameras | main | resolution | $112 \times 112$ |
|---|---|---|---|---|
| | | | field of view | 90° |
| | | | position | 125cm above ground |
| | | | orientation | straight |
| | actions | forward | 25cm | |
| | | turn L/R | 10° | |
| | collision | radius | 10cm | |
| | | height | 150cm | |
| | | sliding | disabled | |
| | cycles | # | per island | 2 |
| | | len | 200 frames (aka. steps) | |
| **NAV-train** | scenes | set | `HM3D/train` | |
| | | # | 800 | |
| | episodes | # | per scene | 10k |
| | | | total | 8M |
| **NAV-test** | scenes | set | `hm3d/val` | |
| | | # | 100 | |
| | episodes | # | per scene | 20 |
| | | | total | 2k |

Table 7: **Parameters for MemNav data generation.**

| Hyper-parameter | Value |
|---|---|
| Sequences | Length ($T$): 100
Random sequence sub-sampling ($T_{\text{sub}}$): 50
Random interval dropping: 8 |
| Batch size | 32 |
| Training steps | 250K |
| Optimizer | Type: AdamW
Weight decay: $5e-2$
$(\beta_1, \beta_2)$: $(0.9, 0.99)$ |
| Learning rate | Min: $1e-8$
Max: $1.5 \times 10^{-4} \times$batch-size$/256$
Warm-up: Linear, for first 20% of iterations
Schedule: Cosine |
| Gradient clipping | Over all parameters, max magnitude:1 |
| Data type | AMP with float16 |

Table 8: **Hyper-parameters used for training models.**

- the choices we made for *Kinaema* worked best for *Kinaema*.

- the choices we made for the baselines, worked best for the baselines.

The *Kinaema* decoder employs a cross-attention (CA) layer with no residual connection whose keys and values are provided by the $Read$ function and queries are the patch tokens from the goal image encoder $Enc_{goal}$. The motivation behind this design is to make the decoder rely solely on the memory outputs and prevent information leakage from $Enc_{goal}$ into the decoder. A learnable token (*i.e.* a CLS token) is attached to the output of the CA layer and given as input to a sequence of 4 standard self-attention (SA) blocks [59]. Finally, the CLS token is detached and relative pose of the goal image is estimated by a MLP with 1 hidden layer.

Other competitive variants, *i.e.* GRU and xLSTM, perform better when multiple CA layers are interleaved with the MLP and SA layers. More concretely, for those models, we attach the CLS token to $Enc_{goal}$ outputs, and apply 3 chains of CA-MLP-SA-MLP layers, where the first CA layer does not have a residual connection. Relative pose estimations are made similarly to Kinaema.

All decoders have a comparable number of parameters.

### C.3 Masked image modeling modules

These modules provide the prediction for the masked image modeling loss, and they follow the design of the RPE modules from the previous section. For Kinaema, a separate sequence of 4 standard SA blocks is applied after the CA layer between $Enc_{goal}$ and $Read$ outputs. Then RGB values for each pixel of a patch are predicted by a linear layer.

On the other hand, for the GRU and xLSTM models, a learned `MASK` token is inserted at the output of $Enc_{goal}$ to the locations of the masked tokens, and they are given as input a separate stack of CA-MLP-SA-MLP layers, followed by a linear layer for predicting RGB values.

## D   Baselines: architecture details

### D.1   GRU

As mentioned in Sec. 5 of the main paper, this variant differs from Kinaema in its $Update$ and $Read$ functions. We implemented the $Update$ function for the GRU variant such that encoded visual and odometry inputs ($\tilde{\mathbf{x}}_t$ and $\tilde{\mathbf{u}}_t$, respectively) are concatenated in the channel dimension and then given to a GRU with 4 hidden layers and 3072 units (following the standard implementation in PyTorch). Then the output from the final step of the GRU is given as input to the $Read$ function, which applies a list of 50 MLPs in parallel to produce 50 different memory tokens. Finally, the memory tokens are given to the $Dec$ functions to estimate the relative pose and model the masked image. The hyper-parameters of this variant, *i.e.* the number of hidden layers and units in GRU and the number of parallel MLPs are chosen such that the total number of trainable parameters of this variant is comparable to that of Kinaema. We saw in general that the more the parameters the better the performance on **RPE-val**.

### D.2   EMA

The EMA models are simple and do not leave much space for configuration choices. The memory vector $\mathbf{m}_t$ is of size 153.6k, chosen to be a multiple of 384. This choice was optimized over the set {19.2k, 38.4k, 76.8k, 153.6k } on **RPE-val**. The visual and odometry encoders $Enc_{vis}$ and $Enc_{odo}$, respectively, are the same as for *Kinaema* and the other baselines. However, the concatenated encoded inputs $[\tilde{\mathbf{x}}_t, \tilde{\mathbf{u}}_t]$ are additionally projected to dimension 153.6k with a single linear layer into, and added to $\mathbf{m}_t$ by simple summing, as was also described in the main paper:

$$Update \triangleq \mathbf{m}_t = \lambda \mathbf{m}_{t-1} + \mathbf{U}\mathbf{x}_t \tag{10}$$

The matrix $\mathbf{U}$ of this layer alone has $(4096 + 64) \times 153.6k = \sim 640M$ parameters. The EMA model trades the simplicity of the update function for the complexity in the input projectors.

### D.3   xLSTM

This variant closely follows the GRU one described above, except that xLSTM[1:0] (comprised only of mLSTM blocks [5]) is used in the $Read$ function. We took the official code [4] for the implementation of mLSTM. To match the trainable parameters of Kinaema, we use 6 mLSTM blocks in total, which are applied sequentially. In each block, the upsampling factor of $1.47$ is used to project the concatenated visual and odometry inputs into $2\times3072$ dimensions, which produces a cell state of size $768\times768\times4 = 2,359,296$. Similarly to the GRU variant, we take the final hidden state (Eq. 21 in [5]) and provide it to the $Read$ function.

### D.4   MooG

We re-implemented MooG with the architecture described in their original paper, but adapted it to our task by giving it the same inputs (including $\mathbf{u}_t$) as the other models. Input images and encoders $Enc_{vis}$ and $Enc_{odo}$ are the same as for *Kinaema* and the other baselines. The rest of the architecture has been kept as in the original paper (ref. [52] in the main paper). Memory $\mathbf{m}_t$ is composed of 1024 embeddings of size 512. As in the original paper, inputs are dealt with patch-wise, in our case the DINO-v2 patch embeddings. We adapt this be concatenating it with the encoded odometry and then

---

[4] https://github.com/NX-AI/xlstm

linearly projecting to the memory embedding dimension of 512. Updates separate a prediction and a correction step.

**Update/prediction step:** as in the original paper, this is implemented as a transformer with 3 layers and 4 heads and self-attention layers.

**Update/correction step:** as in the original paper, this is implemented as a transformer with 2 layers and 8 heads and cross-attention and self-attention layers.

### D.5 Truncated History

We implemented a simple *Truncated History* (Trunc. Hist.) baseline: it directly forwards the concatenated patches embeddings and odometry embeddings to the $Dec$ function.

$$
\begin{aligned}
\mathbf{m}_t &= Update(\mathbf{m}_{t-1}, \tilde{\mathbf{x}}_t, \tilde{\mathbf{u}}_t) \triangleq \begin{bmatrix} \mathbf{m}_{t-1} & \begin{matrix} \tilde{\mathbf{x}}_t \\ \tilde{\mathbf{u}}_t \end{matrix} \end{bmatrix} \\
\mathbf{y}_t &= Read(\mathbf{m}_t) \qquad\qquad \triangleq \mathbf{m}_t
\end{aligned}
\tag{11}
$$

First value is repeated as much as necessary to pad the resulting $\mathbf{m}_t$ matrix to a fixed $T_{\text{trunc}}$ size, eg:

$$
\mathbf{m}_3 = \begin{bmatrix} \underbrace{\begin{matrix} \tilde{\mathbf{x}}_0 & \cdots & \tilde{\mathbf{x}}_0 \\ \tilde{\mathbf{u}}_0 & \cdots & \tilde{\mathbf{u}}_0 \end{matrix}}_{\times T_{\text{trunc}}-3} & \begin{matrix} \tilde{\mathbf{x}}_1 & \tilde{\mathbf{x}}_2 & \tilde{\mathbf{x}}_3 \\ \tilde{\mathbf{u}}_1 & \tilde{\mathbf{u}}_2 & \tilde{\mathbf{u}}_3 \end{matrix} \end{bmatrix}
$$

A maximum of $T_{\text{trunc}} = 100$ embeddings is kept (older ones are discarded), eg:

$$
\mathbf{m}_{243} = \begin{bmatrix} \tilde{\mathbf{x}}_{144} & \tilde{\mathbf{x}}_{145} & \cdots & \tilde{\mathbf{x}}_{242} & \tilde{\mathbf{x}}_{243} \\ \tilde{\mathbf{u}}_{144} & \tilde{\mathbf{u}}_{145} & \cdots & \tilde{\mathbf{u}}_{242} & \tilde{\mathbf{u}}_{243} \end{bmatrix}
$$

### D.6 Integration into the navigation agent

As shown on Fig. 3, we integrate our Kinaema"memory encoder" into the standard agent architecture of [9], which borrows from existing agent baselines models: separate encoders per observation modality, whose outputs are concatenated to be fed to a GRU which maintains an internal state representation (ie. compressed representation of observations history) from step to step while the agent is navigating. This internal state is then given to a linear actor (ie. policy) head to produce a distribution over actions. The originality in [9] is a dedicated binocular encoder pre-trained to compare pairs of images, and estimating relative pose and visibility information between them. This is exploited to predict the relative pose between goal image and an onboard image observation. We extend on this idea by integrating the comparison of the goal to a dedicated compressed memory representation, instead of only comparing to last observation and leaving all the recurrent work to the agent GRU and policy.

## E   Additional experiments

### E.1   EMA w. constant $\lambda$ vs. trainable $\lambda$

In Table 9 we have explored different variants of the EMA baseline with different values of the $\lambda$ decay factor: $\lambda = 0.9$, $\lambda = 0.95$, and a trainable $\lambda$ vector (which is the variant shown in the experiments of the main paper). Making $\lambda$ trainable has a positive impact, which can be explained quite easily: we conjecture that it allows the model to choose whether certain memory features are "fast" or "slow".

### E.2   Alternative read outs for the GRU Baseline

In the main paper we had promised further experiments with alternative read outs for the GRU baseline. Unfortunately we seem to have lost the corresponding model checkpoints, we apologize for that. We are re-running these experiments and will have them ready for the rebuttal period.

| | Seq len 200 | | | Seq len 800 | | |
|---|---|---|---|---|---|---|
| | $1m$ $10^o$ | $1m$ $90^o$ | $2m$ $90^o$ | $1m$ $10^o$ | $1m$ $90^o$ | $2m$ $90^o$ |
| $\lambda = 0.9$ | 4 | 14 | 29 | 2 | 8 | 19 |
| $\lambda = 0.95$ | 2 | 9 | 22 | 1 | 6 | 15 |
| $\lambda$ = trainable vector of size $153.6k$ | 6 | 18 | 34 | 3 | 11 | 24 |

Table 9: **EMA: comparisons of $\lambda$ configurations**, impact on Mem-RPE ( RPE-val ).

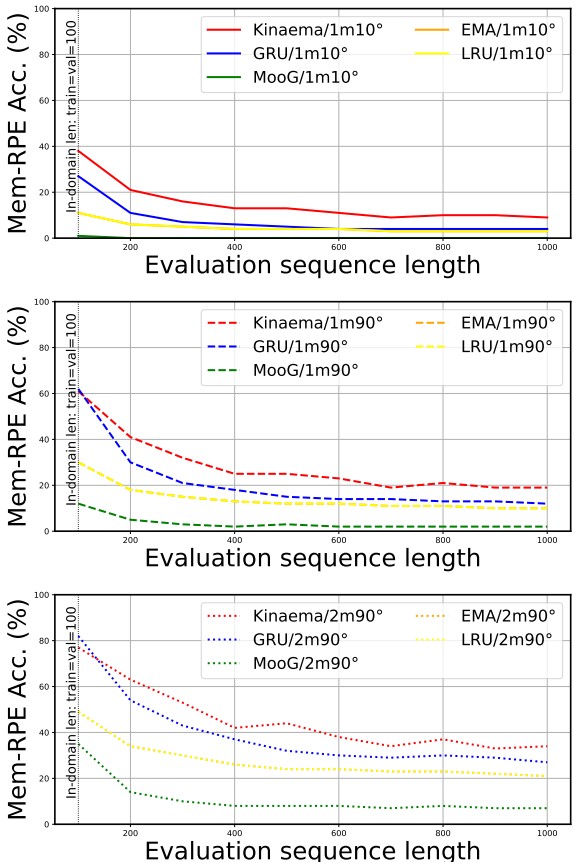

Figure 8: **Comparing all models (trained on $T$=100) on their generalization to longer sequences;** Mem-RPE, RPE-test . (a) pose threshold 1m10°; (b) pose threshold 1m90°; (c) pose threshold 2m90°.

### E.3 Generalization to long sequences

As an extension of Figure 3 of the main paper, which compared the *Kinaema* and GRU models on longer sequences, in Figure 8 of this supplementary material we compare *Kinaema* on RPE-test with all main baselines: GRU, xLSTM, EMA, and MooG. We can see that *Kinaema* clearly outperforms the baselines and provides the best *Mem-RPE* predictions over an extremely large range of out-of-distribution sequences lengths, from the in-domain length of 100 frames up to 1000 frames. The advantage of *Kinaema* is also maintained over all 3 metrics.

### E.4 Training on different sequence lengths

We trained two models on different sequence lengths $T$ and compare their generalization in Fig. 9. The standard model trained on $T$=100 is compared to a variant which has been finetuned on sequences of length $T$=200 (but limiting backpropagation to the last $T$=100 frames of the sequence). In both cases, we provide the *maximum length*, as for each batch we randomly sample sequence lengths. We

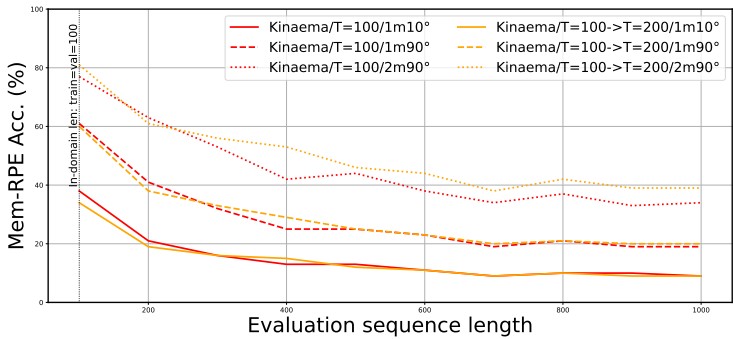

Figure 9: **Impact of different training sequence lengths:** the standard model trained on $T{=}100$ compared to a variant which has been finetuned on sequences of length $T{=}200$. We can see an improvement in precision (the hardest metric) but not necessarily in the overall recall (less hard metrics).

can see an improvement in precision (the hardest metric) but not necessarily in the overall recall (less hard metrics).

### E.5 Visualization of downstream Mem-Nav performances

Tab. 5 provides a plot of the results shown in Tab. 5 in the main paper.

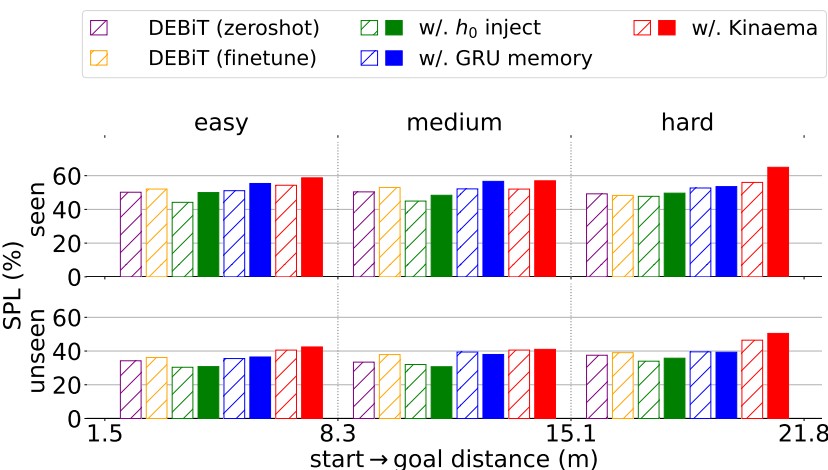

Figure 10: **Navigation efficiency (SPL)** for different episode difficulties (start→goal distance), w/ (■) and w/o (⊟) priming sequence. Numbers correspond to Tab. 5 in main paper.

## F   Occupancy probing experiments

We generate a dataset $\{(\mathbf{m}_t^i, \mathbf{M}_t^i)\}_{i=1...D}$ of $D{=}185$k trajectories of length 100, where $\mathbf{m}_t^i$ are memory states and $\mathbf{M}_t^i$ are corresponding ego-centric 2D metric occupancy maps of size 10m×10m calculated in simulation. A probing network $\phi$, inspired by [64, 10], is trained on Gibson training scenes to predict $\tilde{\mathbf{M}}_t^i = \phi(\mathbf{m}_t^i)$ minimizing the Dice loss [41] between $\tilde{\mathbf{M}}_t^i$ and $\mathbf{M}_t^i$.

The network $\phi$ processes each flattened $\mathbf{m}_t^i$ with an MLP with 2 hidden layers of size 512 to produce an output vector of dimension 2304. This vector is reshaped into a 3D tensor of size $[16, 12, 12]$ and processed by a Coordinate Convolution (*CoordConv*) layer [37], followed by four *CoordConv*-*CoordUpConv* (Coordinate Up-Convolution) blocks. Each such block is composed of:

 **A 2D Dropout layer** with dropout probability $0.05$;

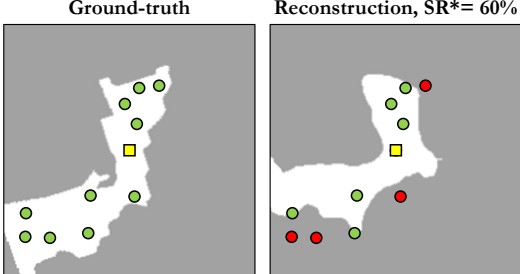

Figure 11: **Computation of *2D Navigation Success Rate*, SR\*:** 10 goal positions $g_n$, in green, are randomly sampled on the ground-truth occupancy map on the left. They are all reachable from the agent position at the center (yellow square). SR\* is computed as the percentage of goal positions $g_n$ that can be reached on the reconstructed map on the right, again in green. Unreachable locations are highlighted in red.

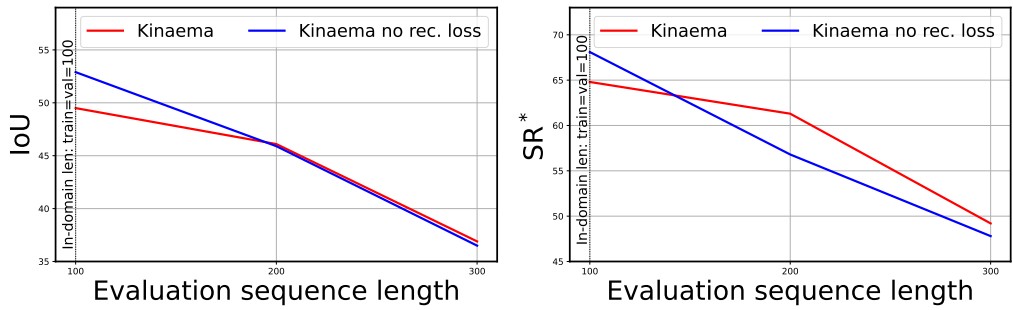

Figure 12: **Comparing occupancy probing performance of Kinaema trained with and without reconstruction loss** with $T{=}100$, on sequences of length $T{=}100, 200, 300$. (Left) Average Intersection over Union, IoU; (Right) Average 2D Navigation Success Rate, SR\*.

**A CoordUpConv layer** with kernel size = 3, stride = 2, padding = 0, that maintains the channel dimension and roughly doubles the spatial dimensions of the feature map;

**A CoordConv layer** with kernel size = 3, stride = 1, padding = 0, that halves the channel size while roughly keeping the other dimensions intact;

**ReLU activation** except for the last block where it is removed.

The result of this process is passed to a Conv2D layer to create the output of size [1, 200, 200] representing the unnormalized logits of each map pixel being navigable.

We tested probing performance on 1000 trajectories of length 100, 200 and 300 collected on unseen Gibson validation scenes, evaluating map reconstruction quality with Intersection over Union (IoU) and assessing how useful the probed map is for navigation with *2D Navigation Success Rate* (SR\*) [10]. Fig. 11 shows an example of SR\* calculation: given a pair of ground-truth and predicted maps $(\mathbf{M}_t^i, \tilde{\mathbf{M}}_t^i)$, we sample 10 reachable goal locations on the ground-truth map $g_n$, and compute SR\* as the percentage of goals $g_n$ that can be reached *on the reconstructed map* $\tilde{\mathbf{M}}_t^i$. See

Fig. 12 shows average probing performance of Kinaema trained with and without reconstruction loss with $T{=}100$, on sequences of length 100, 200 and 300. On the left we display IoU and on the right SR\*. Interestingly, when tested in-domain ($T{=}100$), the model without reconstruction loss achieves better performance, while for longer sequences the trend reverses, especially concerning SR\*. This confirms previous observations that the use of reconstruction loss mainly helps the model generalize to longer sequences.

Finally, in Fig. 13 we display example predictions of occupancy maps obtained by probing the two Kinaema models trained with and without reconstruction loss for sequences of length 100 to 300.

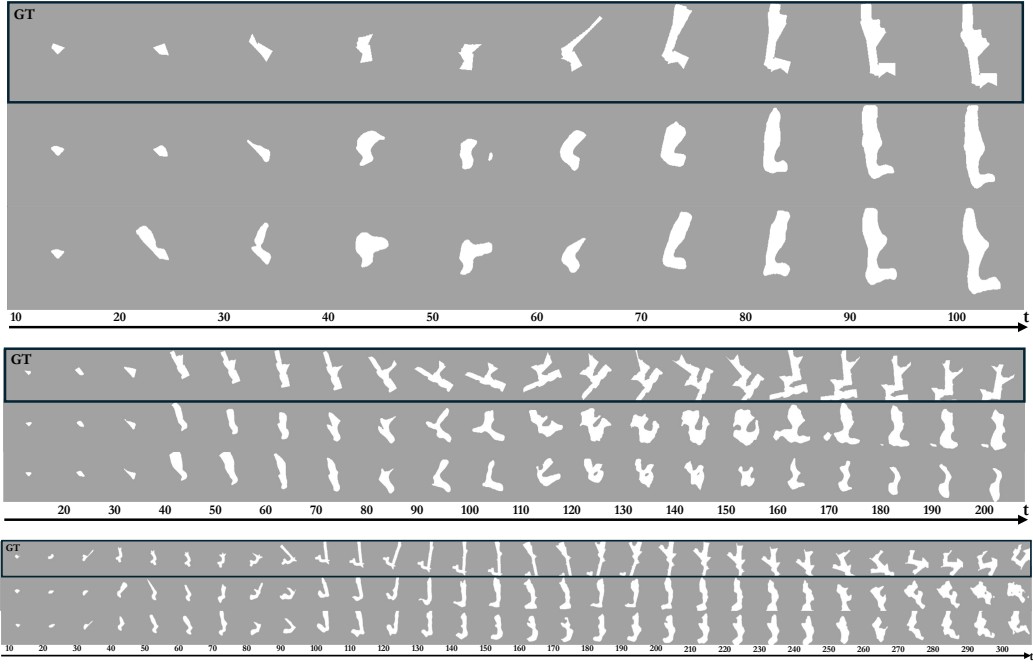

Figure 13: **Probing** of sequences of length $T=100$ (top), $T=200$ (middle), $T=300$ (bottom). On each figure we display, from top to bottom, ground-truth occupancy map, occupancy reconstructed by probing Kinaema, and occupancy reconstructed by Kinaema without reconstruction loss. Sequences are down-sampled by a factor 10.

## G    Additional complementary visualizations of memory attention

### G.1    Attention over top-down maps

In Fig. 14 we show complementary visualizations of memory attention. While in Fig. 6 of the main paper we link patches of observed images to queried memory, in Fig. 14 take a single memory $\mathbf{m}_t$ at time $t$, read out into embeddings $\mathbf{y}_t$, and check how these embeddings are attended given different goal poses. *Kinaema* is primed with the priming sequence shown in green. Then, goal poses are sampled from positions on a regular grid and four different canonical orientations. Respective icons are color coded, where color represents the choice of read out embedding in $\{\mathbf{y}_t\}$ which received highest attention. Figures Fig. 14 (Left) and Fig. 14 (Right) differ in the order in which the priming frames have been visited.

We can see that there are regions of homogeneous color (which identifies the read out embedding) in the map, but also that goal camera orientation potentially plays an important role. This makes sense, as a rotation of the goal camera by $90°$, $180°$ or $270°$ will lead to completely different observations. On the other hand, we see quite big differences between the left and the right Figure, which differ by the order in which the priming frames have been seen. This gives evidence that memory embeddings are not plainly selected based on appearance features, a choice which we would have judged as being sub-optimal, as it would have ignored the spatial layout of a visited scene. As a summary, considering this post-hoc analysis, we judge as positively interesting, that (1) assignments between memory embeddings and goal pose depend on the priming sequence, and (2) there are spatial regularities in highly attended memory embeddings. Combined, they provide evidence for a spatially coherent strategy which does not collapse to appearance alone.

### G.2    Global attention patterns

To assess the contribution of memory tokens across diverse goal images, we compute the cross-attention between the patch tokens output by $Enc_{goal}$ and memory tokens for all 1000 images in

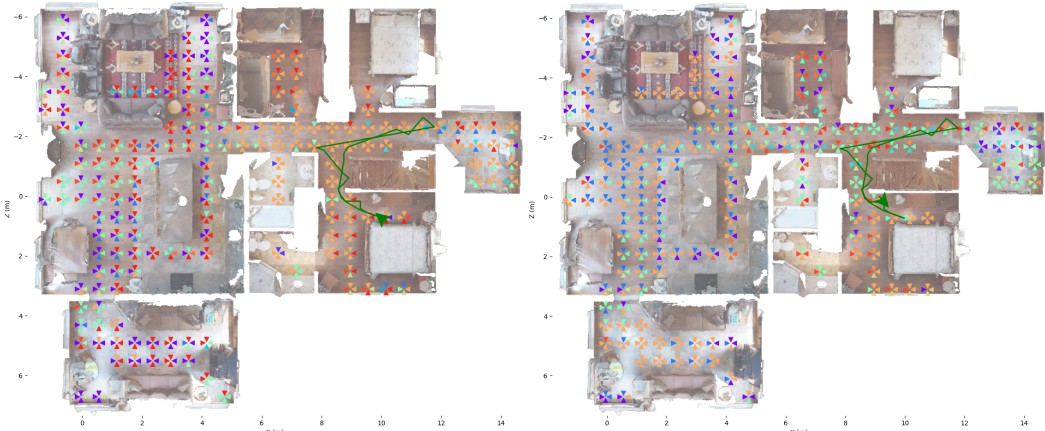

Figure 14: **Visualization of attention to memory queried from different goal poses:** *Kinaema* is primed with the priming sequence shown in green. Then, goal poses are sampled from positions on a regular and 4 canonical orientations, color coded as the choice of read out embedding in $\{\mathbf{y}_t\}$ which received highest attention. Left and right differ in the order in which the priming frames have been visited.

the 200-frame sequences of **RPE-val** . Then we average these probabilities over images and patch tokens, and visualize them in Fig. 15, both per attention head and aggregated across heads.

It can be seen that the heads exhibit distinct and complementary patterns. Heads 2 and 6 concentrate sharply on a few memory tokens, whereas other heads distribute their attention more evenly. Collectively, the heads provide full coverage: the mean attention over all heads assigns non-zero weight to every memory token, although certain tokens are consistently emphasized more than others.

## H    Computing resources and runtime

**Model training** is performed on a variety of NVIDIA V100, A100 or H100 type of GPUs (up to 4 GPUs). To maintain a fixed batch size across all models, we implemented the gradient accumulation mechanism. This way, the batch size per GPU can be as low as 1.

**Mem-RPE** — Training *Kinaema* on 1 H100 GPU takes approximately 5 days.

**Mem-Nav** — RL fine-tuning of the agent w/ *Kinaema* on 1 H100 GPU takes approximately 30 days to reach 300M steps.

**Inference of the full agent** is lightning fast: processing a full set of 2k episodes of **NAV-test** takes approximately an hour and a half on 1 V100 GPU, which corresponds to roughly 140 frames per second.

## I    Broader impact

Our paper investigates the scientific question, whether relative pose can be estimated between an image and latent memory, and whether recurrent Transformers can perform this task without requiring to store historical information. Beyond the exciting scientific reasons for exploring work on spatial AI and embodied AI, we welcome the potentially high interest for society in getting tedious tasks automated. Our contributions advance robotics, and as such, positive and negative impacts are common with other scientific advancements in this area. We can provide a few examples, but the impact of robotics is well known: autonomous agents navigating in indoor spaces could be helpful in health care, care for the elderly, but could also be useful for tasks as mundane as guides in offices, museums *etc.* An intelligent agent could act as an "embodied ChatGPT", providing help not only in the form of textual output, but more importantly, by guiding people to places they have difficulty finding or by fetching items. We acknowledge that any progress in robotics can inherently be abused and produce potential harm, for instance, in surveillance or military applications.

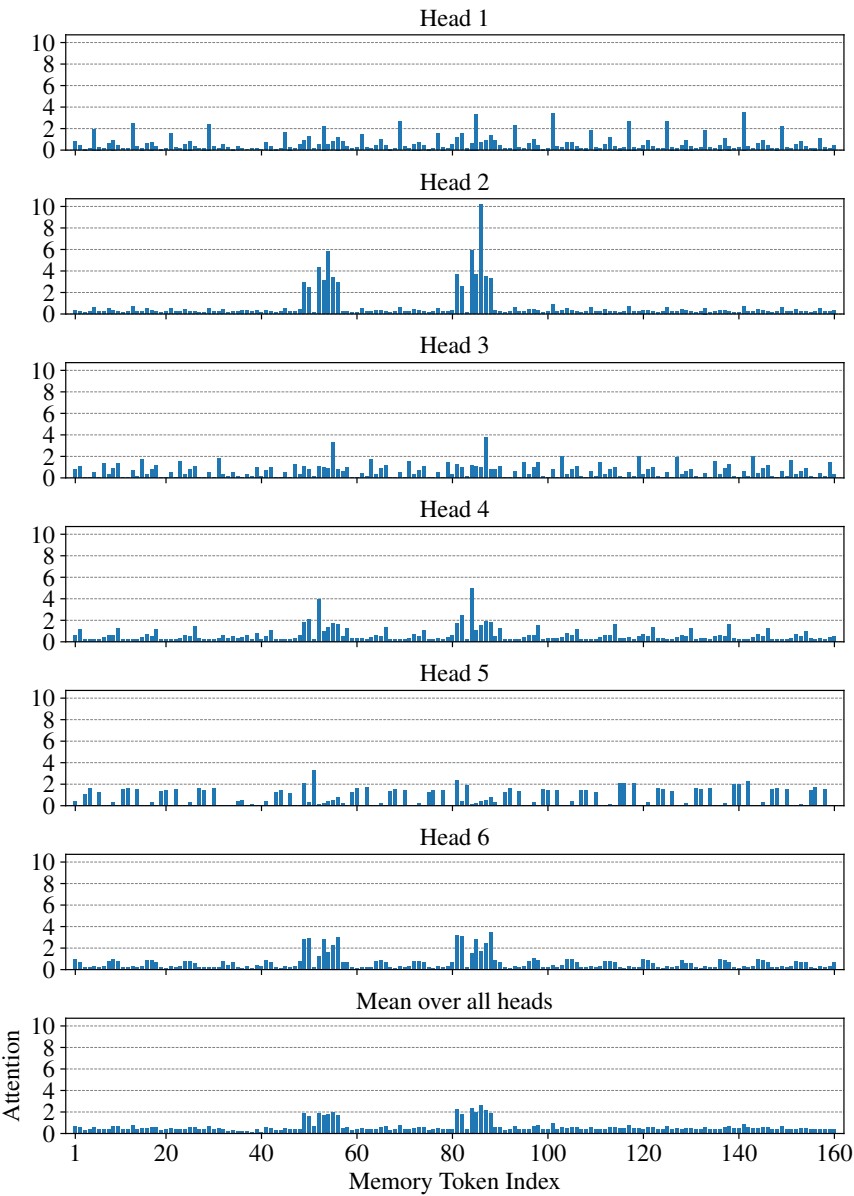

Figure 15: **Cross-attention scores between goal images and memory tokens.** Y-axis represents the attention probability multiplied by 100 for the sake of visualization. Scores for each head sum to 100.

