# OpenReview forum: "Kinaema: a recurrent sequence model for memory and pose in motion"
_NeurIPS.cc/2025/Conference — NeurIPS 2025 poster_

### Official Review · Reviewer_7rrc · 2025-06-24

**Clarity:** 2
**Significance:** 3
**Originality:** 3
**Rating:** 4
**Confidence:** 3

**Summary:**

The authors introduce a new recurrent model termed Kinaema for the Mem-Nav task where the agent first explores the environment and then navigates to the target given a query image. Compared to the quadratic complexity of attention O(N^2), Kinaema is designed to update and query memory in O(1). In addition, gating functions are integrated into Kinaema to decide the speed of updates for each memory item.

Experiments comparing Kinaema with several recent works and classic RNNs demonstrate its advantages in the Mem-Nav task especially for long-term navigation. Ablation studies and visualizations are provided with new insights.

**Questions:**

__Q1__: Line 60: “(iii) the integration of the sequence model in a navigation agent trained with Reinforcement Learning;” It would be great if the authors can explain where the Reinforcement Learning (RL) part is in the paper.

Except for Mem-Nav that looks like the hard version of ε-greedy (__W1__) which relates to RL, the main training is about minimizing the discrepancy of the relative pose between a query image and the current agent’s position, denoted as L_{RPE} on Line 199. However, this formulation does not explicitly relate to the basic framework of RL that involving reward, action, and state. In particular, the paper does not describe a reward function, therefore it is unclear how the agent is “trained with Reinforcement Learning” as claimed in the (iii) contribution.

__Q2__: Line 110 “as p = {t, R}, where t = {d, \theta} is the translation, i.e. distance and bearing angle from the agent to the position depicted in query image. R is the rotation matrix of the goal towards the agent, which is of limited relevance to a navigation task, but which we supervise to increase the learning signal during training.”

There are two rotation-related notations of R and \theta, how are they related to each other? Why is only R used for training but not the other one?

__Q3__: What kind of gating functions are used (Line 145)? Does it follow the standard gating function implementation in the literation? Is it Sigmoid or Tanh? It might be beneficial to add such details for completeness.

__Q4__: What does each color represent specifically in Figure 7?

**Ethical Concerns:**

["NO or VERY MINOR ethics concerns only"]

**Final Justification:**

The authors have provided further details and clarity regarding the relationship of Mem-Nav and ε-greedy, the PPO in the context of RL as well as the Gating mechanism used in this work. In addition, new experiments in the rebuttal demonstrate the advantages over the standard DEBiT. Therefore, I raise my score to 4: Borderline accept.

However, the idea appears to be a slight modification of a recurrent sequence transformer-based model applied to the Mem-Image navigation task. Since the main advantages were demonstrated only in new experiments provided during the rebuttal, and not in the original submission, I believe a score of 4: Borderline Accept is reasonable instead of a higher score.

**Limitations:**

Yes.

**Paper Formatting Concerns:**

Overall, the paper formatting looks good. It would be great to improve the tables and figures listed in __W3,4,7,8__.

**Quality:**

3

**Strengths And Weaknesses:**

__Strengths__:

__S1__: A new recurrent model is introduced with complexity of O(1) to improve classic RNN and attention’s limitations.

__S2__: Figures and tables are presented in high quality.

__S3__: Experiments are comprehensive including comparison with several recent methods an classic RNN baselines.

__S4__: It is good to demonstrate that Kinaema outperforms GRU in each setting.

__S5__: It is good to include experiments with longer trajectories T=200 T=800 to show the memory’s advantages.

__S6__: The appendix is provided with great detail and includes a demo.


__Weakness__:

__P__: Overall, the presentation could be further improved, including __W2-4,7,8__.

__W1__: Differences between Mem-Nav and “Hard” ε-greedy.

The description of “Mem-Nav” from Line 36-38 sounds like a hard version of the classic ε-greedy algorithm in the Reinforcement Learning [1] textbook:

"The method that chooses randomly among all the actions with equal probability ε, and greedily with probability 1 − ε, is called ε-greedy."

Since this work falls within the scope of deep reinforcement learning, I believe it would be beneficial to more clearly differentiate the task from the ε-greedy framework or to discuss the conceptual connections between them.

[1] Sutton et al. Reinforcement Learning: An Introduction.

__W2__: Line 58 “(ii) “Mem-RPE”, a new task requiring the estimation of relative pose between an image and agent memory; “

The term “RPE” appears here for the first time without an explicit interpretation. Providing a clear explanation of the abbreviation “RPE” at its first mention would improve the clarity of the writing.

Line 52-53 in the Appendix provides a good example: “As mentioned in Sec. 4.1 of the main paper, we train all models by jointly optimizing the relative pose estimation (RPE)”.

__W3.1__: The presentation of Table 1 – 4  can be improved. To be concrete, the numbers are left without explanation. Table 5 and Figure 4 are good examples.

__W3.2__: What do the values in green in Table 1 represent? Line 236 mentions two metrics of Success Rate (SR) and SPL, but neither is explicitly referenced in the text or table caption. If the values correspond to Success Rate, it would be helpful to include a percentage sign (“%”), as the numbers appear to range from 0 to 100. Without such clarification or indication of scale, it is difficult for readers to interpret the results. A similar issue is present in Table 4 in the Appendix.

__W3.3__: “SPL” appears for the first time on Line 238 without explanation. Although it is defined in the referenced literature, it would be helpful to also provide the full name within this work for clarity.

__W4__: In Figure 3, the Y-axis is labeled “Mem-RPE Acc. (%)”, but the abbreviation “Acc.” is never used in the main text. Instead, only “Success Rate (SR)” and “SPL” are introduced in Lines 236–238 under the Metrics / Mem-Nav section. It would be helpful to maintain consistent terminology throughout. Similar inconsistencies are also found in Figures 2 and 3 in the Appendix.

__W5__: Figure 4 does not show clear advantages over the standard DEBiT without the introduced modules. It would be helpful to include a discussion or provide new insights to clarify the benefits.

 __W6__: It would be great to provide further discussions of the poorer performance “0 11 22” in Seq len 800 in the last row in Table 2 (a).

__W7__: SR and SPL are reported separately, rather than presented together, in Tables 1–4 and Figures 3 and 4 (Assuming Tables 1-4 represent SR results). It would be more comprehensive to report both metrics side by side, in line with common practice in the literature.

__W8__: It is good to include attention visualizations in Figure 6, but they are difficult to interpret without additional context. For instance, appending a goal image could help assess whether the token attention is meaningful. Further explanation of the color coding would also be helpful.


__Minor Issues__:

__M__: “but using a GRU; (with h0-injection) is a baseline using the hidden state ht of the original agent to encode the priming sequence with no goal, teacher forced,”

The notation of “;” looks weird between “GPU” and “(with h0-injection)”.

---

> ### Author Rebuttal · Authors · 2025-07-30
>
> We thank the reviewer for appreciating the `novelty` (S1), the `quality` (S2,6), and the `relevance` (S3-5) of our contribution. Given the length of this third answer, we formatted it differently from the answers to reviewers rguE and 3MFD.
>
> ###  (W1,Q1) Differences between Mem-Nav and ε-greedy, role of RL.
>
> We do not see a clear link between the Mem-Nav task and the ε-greedy algorithm.
> When we said “the agent can explore the scene” to describe one application case in our introduction, we did not mean “exploration” in the same sense as in the ε-greedy algorithm found in classical Reinforcement Learning (RL) literature.
> ε-greedy policies provide a way to address the classical problem in any optimization of exploitation vs. exploration during the training of the agent with RL. It is a way to explore the state-action space to get better value estimates and avoid local optima to find better policies.
> In our Mem-Nav task, the initial sequence of images provided to the agent at the beginning of a navigation episode that we call priming sequence, or primer, is not meant to help optimize its policy, but to build a better representation of its environment. Letting the agent itself “explore” the environment for a few steps without a goal  was only an application example, such a primer could also come from previous experiences in the same environment, from another agent, solving another task, etc.
>
> In this paper, RL is only involved in the final phase of our training pipeline, the downstream navigation task we call _Mem-Nav_. The main focus of this paper is on the initial pre-training phase, the pretext task we call _Mem-RPE_, which is supervised.
> We apologize for having delegated most of the task definition, the environment model, and the training algorithm to existing work in the literature for this final RL phase. We will add the following details to the paper:
>
> As in the standard ImageNav benchmark [8], for each Mem-Nav episode, the agent is positioned at a random start pose, in a random scene (eg. an apartment), and needs to reach a random goal position.
> The agent receives a single image taken from the goal position, and needs to navigate there using 4 discrete actions: stop, move forward 25cm, and turn left or right 10$^\circ$, based on images observed from the RGB camera mounted on it. Specific to our new _Mem-Nav_ task, the agent also has access to a static sequence of images taken offline, called primer, or priming sequence.
>
> As in [8] and prior work in Embodied AI, the agent receives a dense reward $ r = \Delta d_\mathrm{goal} + 10. \mathbb{1}_\mathrm{success} - 0.01 $, which guides it towards the goal, reinforces success, and encourages shorter paths through slack cost. We train our agent using the Proximal Policy Optimization (PPO) [62] algorithm.
>
> Here the exploration of the state-action space is provided by the PPO algorithm, in particular its entropy regularization term, so there is no need to use an $\epsilon$-greedy policy. Although the primer is collected by the robot in an application sense, in an RL sense, the primer is not actively collected by the agent, it is just part of the initial observation.
>
> In new experiments for this rebuttal, we also ran some form of ε-greedy _evaluation_ of the original DEBiT agent, where we replace the static priming sequence by an “active” exploration phase of same length (200 steps) during which random actions are selected with probability 1/10 (greedily following the trained policy otherwise). Results are given in the table below (cf answer to W5). Values are not directly comparable to the ones with primers which are generated from a heuristic designed to maximize coverage of the scene (see Appendix A.2) but are _goal-less_. ε-greedy exploration might have worse coverage but the trained policy will try to go towards the goal. Nonetheless, results are on par if not lower than the zero-shot baseline, which would indicate that the agent was not able to build useful internal states from these ε-greedy explorations.
>
> [62] Proximal Policy Optimization Algorithms - John Schulman, Filip Wolski, Prafulla Dhariwal, Alec Radford, Oleg Klimov
>
> ### (W5) Advantages over the standard DEBiT
>
> In new experiments, longer fine-tuning (300M steps vs. 100M for the results reported in the initial submission) have revealed a clearer advantage of Kinaema on Mem-Nav task, as shown in the table below:
>
> | SPL (%) | DIST SPLIT → | EASY  | (2-5m) | MEDIUM  | (5-10m) | HARD | (10-22m) |
> |:--- | ---:| --- | --- | --- | --- | --- | --- |
> |  | SEEN SPLIT → | SEEN  | UNSEEN  | SEEN  | UNSEEN  | SEEN  | UNSEEN  |
> | ↓ MODEL | PRIMER ↓ | (#=566) | (#=116) | (#=520) | (#=319) | (#=64) | (#=290) |
> | • DEBiT (zeroshot) | ✘ |   41        |   41        |   40        |   42        |   47         |   41        |
> | • DEBiT (finetune) | ✘ |   45        |   40        |   42        |   43        |   48         |   44        |
> | • DEBiT (ε-greedy) | ✘ |   39        |   34        |   35        |   36        |   38         |   36        |
> | • w/. $h_0$ inject | ✘ |   37        |   32        |   35        |   37        |   42         |   38        |
> |   | ✔ |   41   (+4) |   40   (+8) |   36   (+1) |   38   (+1) |   41   (-1)  |   40   (+2) |
> | • w/. GRU memory | ✘ |   43        |   42        |   42        |   43        |   50         |   45        |
> |  | ✔ |   45   (+2) | **48** (+6) |   43   (+1) |   46   (+3) |   50   (+0)  |   45   (+0) |
> | • w/. Kinaema | ✘ |   48        |   47        |   46        |   45        |   44         |   48        |
> |  |  ✔ | **50** (+2) |   46   (-1) | **48** (+2) | **49** (+4) | **54** (+10) | **49** (+1) |
>
> where all baselines have also benefited from the same extended training,
> including fine-tuning of DEBiT (imagenav) in hm3d/train.
> Parenthesized deltas highlight the capacity of each model to exploit the priming sequences collected offline.
> We also distinguish between episodes where the goal has been SEEN or UNSEEN in the priming sequence, as requested by reviewer 3MFD. We also kindly refer to comments in the answer to this reviewer.
>
> These new numbers will be used to update Figure 4 and enrich the corresponding discussion in the text.
>
> ### (W6) “Poor performance in last line of Table 2”
>
> Tab.2a ablates memory capacity and is commented at L.295-301.  We conjecture that the drop in evaluation over longer (T=800) sequences observed for the maximum capacity of $50$ mem-tokens is caused by a too large memory capacity wrt to the trajectory length, as there is less need for compression — it is widely conjectured that compression is linked to intelligence. For this reason, the compression-less memory handling strategy learned by the model tends to overfit and does not generalize well to longer sequences.
>
> ### (Q3) Gating
>
> We have provided a full explanation of gating with the corresponding equations in the answer to reviewer rguE and we kindly refer to this part for details. In the camera ready, we will update Eq.5 with these details.
>
> ###  (P, W2, W3, W4,W7,W8)
>
> The overall presentation of the paper will be improved as followed:
>
> * Acronyms will be defined at their first appearance:
>     - (W2) l.58:
>
>     > (ii) **R**elative **P**ose **E**stimation from **Mem**ory (Mem-RPE), a new task [...]
>
>     - (W3.3) l.236-238:
>
>     > **S**uccess **R**ate (SR) [...], and **S**uccess weighted by **P**ath **L**ength (SPL) [2], $SPL=$ [...]
>
>     - (W4): l.232:
>
>     > We provide **Acc**uracy (Acc.) of recognized poses [...]
>
>     y-label of Fig.4 replaced by
>
>     > Accuracy (%)
>
> ### (W3.1,3.2,7) Description of Tab.1-4
>
> Metrics used for Mem-RPE evaluation in Tab.1-4 introduced at l.229-233
> will be explicitly reminded in their captions, eg. Tab.1:
>
> > Mem-RPE **Accuracy** for different models, sequence lengths and (distance,angle) thresholds
>
> Improved captions should also address misunderstanding (W7) where _Mem-RPE_ Accuracies
> have been confused with _Mem-Nav_ SR, which is reported in Tab.5 only, alongside SPL.
>
> ### (W8, Q4) Meaning of colors in Fig.6
>
> Assuming (Q4) referred to Fig.6 instead of Fig.7 which is grayscale, we will clarify caption of Fig.6 and the related discussion l.334-343:
>
> In this visualization, each image of the sequence has been used as a goal image (which is normally fixed during a navigation episode) to query memory built up from past observations. The purpose of the figure is to highlight attention patterns and coherence along a sequence, colors are arbitrary and only used to identify which mem tokens "activate" on each patch (no associated semantic).
>
> ### \(M\) Inline format of enumeration
>
> The semicolon ";" denotes separation of items in the enumeration of variants labelled with the text in parentheses, each corresponding to a row in Tab.5
>
> ### (Q2) Confusion between notations $\theta$ and $R$
>
> Sorry for the confusion, we will redefine $\mathbf{t}$ as a 3D translation vector $\{x,y,z\}$ in cartesian coordinates from current agent position to goal position. We had chosen to define it as a 2D vector $\{d,\theta\}$ in polar coordinates, where $d$ is the euclidean distance from agent to goal, and $\theta$ the angle towards the goal from the direction faced by the agent. This initial choice was motivated by its significance and interpretability for navigation in general, where agents evolve mostly in 2D and need an egocentric view of their goal, but is not what we used for Mem-RPE, where we cared about 3D camera poses and image alignment.
> On the other hand, $R$ is a 3D rotation matrix describing the relative orientation of the goal camera wrt. the one of the agent. It is not used in the ImageNav and Mem-Nav tasks, and only provides some supervision signal to implicitly align point of views and compare images for Mem-RPE.
>
> ### (Q4) Meaning of colors in Fig.7
>
> Fig.7 represents top-down occupancy grid where _gray_ is unknown or obstacle and _white_ is free/navigable.

---

> > ### Comment · Reviewer_7rrc · 2025-08-03
> >
> > Thank you for the detailed responses, which addressed some of my concerns. I think adding brief explanations for questions from answers to (P, W2, W3, W4,W7,W8) through (Q4) Meaning of colors in Fig.7 would further improve the paper's presentation.

---

> ### Author Response · Authors · 2025-08-04
> **Response to 7rrc**
>
> Thanks for your response.
> We will indeed integrate these answers into the final version for better clarity. Please let us know if some answers still lack some details or if we can clarify any other point.

---

### Official Review · Reviewer_3MFD · 2025-07-02

**Clarity:** 4
**Significance:** 3
**Originality:** 4
**Rating:** 5
**Confidence:** 4

**Summary:**

This paper introduces Kinaema, a novel high-capacity recurrent sequence model designed for embodied agents that operate in long-horizon, continuous visual navigation tasks. The key innovation lies in maintaining a distributed memory composed of multiple embeddings, updated recurrently via a Transformer block and stabilized using gating mechanisms inspired by GRUs. The authors evaluate Kinaema on two newly proposed tasks: Mem-RPE (estimating relative pose between a query image and the agent’s latent memory) and Mem-Nav (navigating to a previously seen target location, using only prior observations). Empirical results on simulated 3D environments (HM3D, Gibson) demonstrate that Kinaema outperforms multiple recurrent baselines (GRU, xLSTM, MooG) in both memorization and navigation efficiency, especially for long sequences. Ablations support the importance of transformer updates, gating, and distributed memory structures.

**Questions:**

1. Test Time Training introduces fast weights and could also memorized long sequence with O(1) computational complexity. The authour could add some disscussion on using recurrent transformer compared to TTT.

2. Besides using architecture with O(1) computational complexity, the author could also compare to methods merging KV cache as memory in transformer.

**Ethical Concerns:**

["NO or VERY MINOR ethics concerns only"]

**Final Justification:**

From the extra result, I agree that the proposed recurrent sequence model is indeed a solid memory mechanism at the current stage. I would advocate the acceptance of this paper.

**Limitations:**

The potential for **lifelong memory** or **multi-episode navigation** is not addressed. The proposed navigation agent is a modular pipeline and **the memory is not updated during navigation**, making it unsuitable for dynamic scenes or tasks that require continual perception.

**Quality:**

4

**Strengths And Weaknesses:**

## Strengths
1. The paper introduces Mem-RPE and Mem-Nav, two tasks that explicitly target the underexplored and practically important setting of continuous memory-based navigation, aligning more closely with real-world robotics deployment than traditional episodic benchmarks.

2. The Kinaema model is designed to support O(1) memory updates and access, in contrast to the O(N²) cost of standard attention-based transformers. This architectural choice is particularly well-suited for long-horizon tasks where memory scaling is critical.

3. The paper provides a thorough empirical evaluation, including generalization to out-of-distribution sequence lengths, memory scaling studies, and ablation analyses. The inclusion of multiple competitive baselines (GRU, EMA, xLSTM, MooG) allows for a fair and comprehensive comparison.

4. The paper is well-organized and clearly written, with strong exposition of model architecture, training setup, and evaluation procedures. Visualizations such as memory attention maps and cross-attention overlays further aid understanding.

---

## Weaknesses

1. **Streaming history is modeled as random exploration** via varied-length sampled trajectories, rather than through an **active or structured exploration policy**. As a result, there is no analysis of **visual coverage or informativeness** of the memory contents. This limits insights into whether the model benefits from task-relevant scene exploration, which would be expected in real-world deployment.

2. The **model assumes that the goal image comes from a previously observed location**, but the model lacks explicit uncertainty estimation (e.g., recognizing if a goal has not been seen), or out-of-distribution detection.

3. In the **Mem-Nav task**, Kinaema is **used as a frozen plugin module** to DEBiT rather than training a end-to-end navigation policy with recurrent memory. This modular design limits the agent’s capacity that learns to memorize navigation history and learn about the environment lifelong. In another word, **the memory is not updated during navigation**, making it unsuitable for dynamic scenes or tasks that require continual perception.

4. The potential for **lifelong memory** or **multi-episode navigation** is not addressed. Recent work such as **GOAT-Bench** (Khanna et al., CVPR 2024) directly tackles multi-task or long-term memory settings and could provide a more fitting benchmark or discussion context.

5. While comparisons to GRU, xLSTM, EMA, and MooG are helpful, **Mamba (Gu & Dao, 2024)** is notably missing. Mamba’s state-space approach also introduce a distributed memory (D SSM applied to each channel), and would serve as a more **direct architectural comparison** to Kinaema than classical RNNs.

6. Although the model is trained on **sequences of length T=100**, evaluation is extended to sequences of up to T=1000. However, there is **no justification for the limited training horizon**, nor any exploration of **scaling effects** (e.g., whether longer training sequences would improve generalization). The sharp drop in accuracy between T=100 and T=200 suggests potential **brittleness and overfitting** to the training length.

---

> ### Author Rebuttal · Authors · 2025-07-30
>
> We thank Reviewer-3MFD for their review, for `having appreciated the introduction of underexplored new tasks`; the `support of O(1) updates well-suited for long-horizon tasks`; the `thorough empirical evaluation`, `generalization, ablations  and multiple competitive baselines`; the `well-organized and clearly written paper`.
>
>
> > 1. no analysis of visual coverage or informativeness of the memory contents.
>
> We have made new experiments which provide this analysis below, distinguishing between episodes where the goal position has been *SEEN* or *UNSEEN* during the priming sequence, and also whether the priming sequence has actually been presented to Kinaema or not (if not, then SEEN and UNSEEN have the same meaning).
>
> These are downstream Mem-Nav experiments, similar to Figure 4. We also keep the splits of episode difficulty but restricted it to 3 different difficulty levels to not overcharge the table.
>
> | SPL (%)            | DIST SPLIT → | EASY        | (2-5m)      | MEDIUM      | (5-10m)     | HARD         | (10-22m)   |
> | -- | -- | -- | -- | -- | -- | -- | -- |
> |                    | SEEN SPLIT → | SEEN        | UNSEEN      | SEEN        | UNSEEN      | SEEN         | UNSEEN     |
> | ↓ MODEL            |     PRIMER ↓ | (#=566)     | (#=116)     | (#=520)     | (#=319)     | (#=64)       | (#=290)    |
> | • DEBiT (zeroshot) |            ✘ |   41        |   41        |   40        |   42        |   47         |   41        |
> | • DEBiT (finetune) |            ✘ |   45        |   40        |   42        |   43        |   48         |   44        |
> | • w/. $h_0$ inject |            ✘ |   37        |   32        |   35        |   37        |   42         |   38        |
> |                    |            ✔ |   41   (+4) |   40   (+8) |   36   (+1) |   38   (+1) |   41   (-1)  |   40   (+2) |
> | • w/. GRU memory   |            ✘ |   43        |   42        |   42        |   43        |   50         |   45        |
> |                    |            ✔ |   45   (+2) | **48** (+6) |   43   (+1) |   46   (+3) |   50   (+0)  |   45   (+0) |
> | • w/. Kinaema      |            ✘ |   48        |   47        |   46        |   45        |   44         |   48        |
> |                    |            ✔ | **50** (+2) |   46   (-1) | **48** (+2) | **49** (+4) | **54** (+10) | **49** (+1) |
>
>
> Remarks: these new results are better than what we obtained in the submission as we also RL-trained the downstream agent longer (300M steps instead of 100M). All other parts are identical, in particular the pre-trained Kinaema and GRU models.
>
> As we can see,
> * the model is capable of exploiting the additional information provided in the priming sequence and obtains gains: (values provided in parentheses).
> * The gains are largest for the hardest sequences (+10p)
> * The gains are not restricted to “seen” episodes. For the “unseen” ones, while the goal object had not been seen in the priming sequence, Kinaema picks it up when it is seen during the actual navigation (Kinaema memory is updated during navigation) and it does not lose it out of mind when it is potentially briefly not seen anymore, in contrast to the DEBiT encoder, which only compares the current observation to the goal.
>
> We will include these results in the camera ready.
>
> > 2. the model lacks explicit uncertainty estimation (e.g., recognizing if a goal has not been seen)
>
> We acknowledge this as a minor limitation of the base Kinaema model. However, we would like to point out two ways how the agent can circumvent this limitation, and the results above provide evidence for that:
>
> - (1) if Kinaema has not seen the goal position, its prediction will be OOD and the agent will observe this during RL training. Through RL-training, it can potentially detect OOD patterns.
>
> - (2) the agent also has access to the DEBiT encoder, which compares the currently observed image with the goal image and which has been trained for visibility prediction. The agent can therefore learn to trust DEBiT if it reports to have seen the goal, and to trust Kinaema in the contrary (or to detect its OOD behavior).
>
> > 3. the memory is not updated during navigation
>
> There seems to be a misunderstanding, **Kinaema memory IS UPDATED during navigation**, and we believe that this is one of its core strengths . The two memories $\mathbf{m}_t$ and the agent’s own $\mathbf{h}_t$ are **BOTH** updated in parallel. In the integration Figure 1 of the appendix we omitted this update to remove clutter, we apologize for that. We will make this very explicit in the camera-ready.
>
> > 4. The potential for lifelong memory or multi-episode navigation is not addressed. Recent work such as GOAT-Bench (...)
>
> Kinaema has indeed the potential to target multi-episode nav and we actually see our Mem-Nav task as a variant of multi-episode nav: the priming sequence represents the information collection step of a previous episode. There is a difference to GOAT-Bench: during our priming sequence, the agent does not take its own decisions. However, as mentioned in L75, this was a deliberate choice, as it allows us to pre-compute memory representations *before* each RL-episode.
>
> We would like to stress that this is an important advantage  as it **provides a tremendous speed-up during RL training**. During early training phases, agents tend to make mistakes and early episodes are extremely short. Our Mem-Nav task allows the agents to immediately access useful memory even in the early training phases.
>
> We would again stress that pre-computations are only done for the observations collected during the priming sequence. During the subsequent RL steps, Kinaema memory continues to be updated.
>
> Our model could in principle also be applied to the GOAT-Bench, up to the described limitations, and by modifying the GOAT-Bench to deal with image goals in each sub-episode.
>
> > 5. comparisons to GRU, xLSTM, EMA, and MooG are helpful, Mamba is notably missing.
>
> As requested, we have implemented the Mamba baseline, as well as another state space model targeting long sequences, LRU (Orvieto et al., Resurrecting Recurrent Neural Networks for Long Sequences, ICML 2023).
>
> - For LRU, we used the author’s sysid-pytorch-lru repository from forgi86 on github. We have launched multiple runs with different numbers of layers and the same hyperparameters as Kinaema and GRU (emb size=3072, LR, scheduler etc). Training is still in progress due to the short rebuttal time, but we are close to finishing (epoch 147 of 200, and most models do not provide significant gains over the last 30% of training). The training and validation curves are significantly worse than Kinaema.
>
> We provide preliminary evaluation results (Mem-RPE task, RPE-test set) below, which are comparable to the eval protocol in Table 1. We plan to report the full results during the reviewer/author discussion phase.
>
> The LRU results are significantly below the results of our model.
>
> Seq-Len 200: Accuracies 3, 19 and 36 (for the 3 different metrics)
>
> Seq-Len 800: Accuracies 2, 12  and 24
>
> (the best model had 4 layers, compared to 3 layers for Kinaema)
>
> - For Mamba, we tried the state-space repositories both on github and huggingface. However, due to large fluctuations of activations within the mamba modules, we haven’t managed to fully train one so far, despite our efforts to provide additional regularization tailored to the Mamba model and which was not necessary for Kinaema or the other baselines. We will do our best to report the results of a stable training in camera ready.
>
> >  6. trained on sequences of length T=100, evaluation is extended to sequences of up to T=1000.
>
> We kindly refer to Figure 3 of the supplementary material, where we show results training Kinaema (and the GRU baseline) for T=200 steps, which leads to improvements in Mem-RPE performance, in particular in precision (the first metric).
>
> We would also like to point out that the current recurrent transformer models of the **literature** are trained on **significantly shorter sequences**. MooG [52], for instance, has been trained for T=8 and evaluated to seq of length 16.
>
> > TTT introduces fast weights and could also memorized long sequence with O(1) computational complexity.
>
> TTT is a recurrent model as are RNNs, LSTMs, GRUs, MooG, or our model, and as such it shares some of their properties, like $O(1)$ complexity and the compression of the history of observations into some memory. Instead of compressing the history of observations into an embedding (RNN/LSTM/GRU) or a set of embeddings (MooG/Kinaema), the  TTT model compresses it into theparameters of a small memory-network. Memory updates are therefore gradient descent steps.
>
> In direct comparison to Kinaema, TTT obtains $O(1)$ by requiring to do memory updates with gradient descent, which comes with its own challenges. Integrating information into a memory this way depends on learning rates and the number of learning steps for each memory write and is generally sensitive to (inference time) optimization hyperparameters and objectives, the requirements of doing some form of batching even during inference time (where sequence batchsize is 1), and the computational downside of doing backward passes on a model during inference.
>
> > The author could also compare to methods merging KV cache as memory in transformer.
>
> Methods caching KV embeddings can decrease the computational complexity for each prediction from the classical transformer complexity of $O(N^2)$ to $O(N)$. This is achieved through causality (embeddings do not attend to the future, so once they are contextualized, they are not touched anymore and can be cached).
>
> While this provides a tremendous speedup, it is not O(1) complexity, as our model provides. These methods still need to attend to the full observation history at time step for each query, hence the $O(N)$ complexity. Furthermore, it does not remove the necessity to store the full observation history. The context length problem is not solved, compared to our model.

---

> > ### Author Response · Authors · 2025-08-02
> > **Promised updates on LRU baseline experiments**
> >
> > As promised, we provide the updates of the final performance of the state-space model “LRU”, which has finished all 200 epochs (equal to the proposed Kinaema model and the other baselines).
> >
> > There was a very slight increase in performance compared to the intermediate checkpoint results we provided in the rebuttal, but the LRU results remain significantly below the results of our model. These are RPE-test results with the evaluation protocol comparable to the values in Table 1 of the submission.
> >
> > Seq-Len 200: Accuracies 4, 22 and 38 (for the 3 different metrics)
> > Seq-Len 800: Accuracies 3, 14  and 24
> >
> > (the best LRU model had 4 layers, compared to the optimum number of 3 layers for Kinaema. All hyper-parameters are identical to Kinaema or the other baselines)
> >
> > We are happy to answer any additional questions you might have.
> >
> > The "Kinaema" authors

---

> > ### Comment · Reviewer_3MFD · 2025-08-07
> >
> > I appreciate the author's additional experiments on the analysis of memory content, extra baselines of Mamba, and LRU.
> > I agree with the author's discussion on TTT and KV caching memory, and the proposed recurrent sequence model is indeed a solid memory mechanism at the current stage. I would appreciate it if the author could add these discussions to the paper.

---

> > > ### Author Response · Authors · 2025-08-08
> > > **Discussion by Authors**
> > >
> > > Thank you for your response. We’re glad to be aligned on the difference between the memory mechanisms, and we appreciate your recognition of the additional experiments and analysis. We’ll incorporate the discussions and results into the final version for completeness. We hope that our rebuttal and discussion will be reflected in a score increase.

---

> ### Author Response · Authors · 2025-08-05
> **Discussion by Authors**
>
> Dear reviewer, may we kindly ask whether all your questions have been answered? The author / reviewer discussion is drawing to an end and we want to be sure that you do not require any further information.

---

### Official Review · Reviewer_rguE · 2025-07-05

**Clarity:** 3
**Significance:** 2
**Originality:** 3
**Rating:** 3
**Confidence:** 3

**Summary:**

The paper introduces a new model architecture for storing and retrieving long-term memory specifically for the task where a robot needs to estimate the relative pose between a target image and the current pose. While the ideas presented make intuitive sense, I find the writing hard to follow in some cases.

**Questions:**

1. How does the Kinaema model fit into the steps describe in the background in the beginning of section 4

**Ethical Concerns:**

["NO or VERY MINOR ethics concerns only"]

**Limitations:**

please see my comments above

**Quality:**

3

**Strengths And Weaknesses:**

Quality

I like the discussion around the desired behavior of the model and the consideration that go into designing the model architecture at the beginning of section 4. The discussion makes sense and provide a holistic explanation for more technical ideas that follow.

Clarity

On line 144-145, the paper writes this `allowing the model to take decisions (to “gate”) on the speed of updates for each memory item`. What does it mean to take decision on the speed of updates?

On line 172-173, "the paper writes the following self-attention transformer contextualizes the embeddings with each other". Self-attention is an operation. What does it mean to have a self-attention transformer? And what does contextualizes the embeddings with each other mean?

It is also unclear the relationship between the model described in section 4.1 and background information provided in section 4.

Significance

I agree with the argument in the paper that the more traditional navigation setting doesn't lend itself well to study longer memory context problem. However, I would have like more description on the task in the background session to be able to understand the experiment better.

Originality

The experimental setting that the paper studies and the ideas presented are novel to the best of my knowledge. I don't work in navigation however and I'm not very familiar with the literature.

---

> ### Author Rebuttal · Authors · 2025-07-30
>
> We thank Reviewer-rguE for their review, for highlighting that `the ideas presented make intuitive sense`, and for liking `the discussion around the desired behavior of the model and the consideration that go into designing the model architecture`. We provide an answer to the main remaining questions below.
>
>
> > It is also unclear the relationship between the model described in section 4.1 and background information provided in section 4.
> > How does the Kinaema model fit into the steps described in the background in the beginning of section 4.
>
>
> We directly used goals G1, G2, G3 to design the Kinaema model, these goals were indeed the background for our design choices.
>
>
> Most importantly, we want our model to be recurrent (G1), so that one time step is of complexity $O(1)$ (taking into account the number input time steps in the big-O notation) and thus that it scales to long sequences with low complexity, eg. for robotics applications. This is opposed to transformer based solutions with attention over input tokens, as the standard models used for LLMs or LVAs. These existing solutions require storage of the history observations and attend over them, and are of complexity $O(N^2)$, or, if causal and using KV-caching, $O(N)$.
>
>
> Existing *recurrent* models (RNN/GRU/LSTM/…) do indeed already feature this target complexity of $O(1)$, but they lack network capacity, as their number of parameters is quadratic with respect to memory size. Our design goal (G2) is to scale memory size, which is why we extend memory embeddings from a single embedding to multiple embeddings. Together with the model choice, this addresses (G2), as the update function of our recurrent model is a transformer, which scales over memory size (transformer complexity independent of the number of embeddings, which is in our case the number of *memory* embeddings).
>
>
> Lastly, some recurrent models (like classical RNNs) have problems with stability, as the backpropagation chain is complex. This has been addressed by LSTMs and GRUs through gating functions, which better control the speed of individual updates (for more details on this see the answer on gating further below). Stability is our design goal G3, for that reason we take the gating functions from GRUs and adapt them to our transformer-based recurrent model. This required adapting them to the presence of multiple memory embeddings, compared to the model where they originate from, GRUs.
>
>
> **As a summary**, to achieve our 3 goals (G1, G2, G3), we design a transformer-based representation that is *updated recurrently* by a gating mechanism (inspired by classic GRU architectures), such that it can be updated with $O(1)$ and thus scaled to long sequences (G1). Almost all components of our model are based on self- and cross-attention to maintain the capacity and expressive power of modern transformer-based architectures (G2). Finally, we adopt an approach similar to that used in GRUs and LSTMs to address training instability issues of early RNNs, that is, we use a gating mechanism to control the memory update and stabilize training (G3).
>
>
>
> > On line 144-145, the paper writes this allowing the model to take decisions (to “gate”) on the speed of updates for each memory item. What does it mean to take decision on the speed of updates?
>
>
> Gating functions are classically used in sequence models such as LSTM or GRU. Each trainable gating function predicts a single scalar between 0 and 1 and does this *for each embedding dimension*. This gating value is then multiplied with the corresponding feature value. Different gates have different purposes: some gates regulate whether a memory value is updated by the model or whether its old value is kept, or rather a continuous value between these two choices.
>
> As written in the paper, we use the GRU gating functions, which we provide below for completeness (see also Chris Olah’s blog on “Understanding LSTM Networks” for an excellent breakdown including visualizations):
>
>
> $ z_t =\sigma\left(W_z \cdot\left[h_{t-1}, x_t\right]\right) $
>
>
> $ r_t =\sigma\left(W_r \cdot\left[h_{t-1}, x_t\right]\right) $
>
>
> $  \tilde{h}\_t = \tanh (W \cdot  [r_t * h_{t-1},  x_t ] ) $
>
>
> $ h_t =\left(1-z_t\right) * h_{t-1}+z_t * \tilde{h}\_t $
>
>
> For GRUs, the reset gate $r_t$ controls for each dimension of the GRU memory $h_t$ how much of its value is forgotten when we calculate the updated values. The update gate $z_t$ controls for each memory dimension how much we update it (as opposed to keeping the old value).
>
>
> For our proposed Kinaema model (L177-187), we use the same gates as above and the same update functions. However, these updates are done for each of the Kinaema embeddings separately. The GRU equations above apply, with the input $x_t$ corresponding to the “update candidates” $\mathbf{\tilde{m}}\_{t,n}$ produced by the transformer block in equation (4). The matrices $W\_z, W\_r, W$ above are thus shared over Kinaema memory embeddings: the same gating function is re-used for every Kinaema memory embedding.
>
>
> > On line 172-173, "the paper writes the following self-attention transformer contextualizes the embeddings with each other". Self-attention is an operation. What does it mean to have a self-attention transformer? And what does contextualizes the embeddings with each other mean?
>
>
> By self-attention transformer, we meant a standard transformer block, each composed of multi-head self-attention and an MLP; both including with layer norms, and residual connections; however, we wanted to emphasize the type of attention used, ie. “self-attention” and not “cross-attention”, as this would have been an alternative which is used for similar purposes for instance in MooG [52] (or in our memory decoder in Equation (2)). We will make this clearer.
>
>
> To “contextualize an embedding” means to update the embedding features with information from its surrounding context, ie. features from the other embeddings of the sequence or set, so that the resulting representation is aware of the broader input or sequence it belongs to. With this we meant the standard operation done by transformers and we borrowed the term from the NLP literature. We will remove it or explain the concept differently.

---

> ### Author Response · Authors · 2025-08-05
> **Discussion by Authors**
>
> Dear reviewer, may we kindly ask whether all your questions have been answered? The author / reviewer discussion is drawing to an end and we want to be sure that you do not require any further information.

---

### Note · Authors · 2025-08-11

As final comments, keeping them as **short as possible, only the most important info**, let us summarize the content of (A) paper and (B) some answers to reviewers’ concerns:

**(A) Original submission**:
- We propose a new seq model which is both **recurrent** and **transformer-based** and provides $O(1)$ memory updates and high memory capacity.
- We pre-train it for a new vision task for robotics estimating relative pose between a query image and content previously seen in the history of observations.
- We insert the model into a navigation agent which is capable of navigating to previously seen places.

**(B) Rebuttal and additional content for the camera-ready**:
* Kinaema **IS** updated during navigation
* New analysis of visual coverage
* New baselines (state space models)
* New improved results w. longer downstream training (300M steps), pre-training unchanged
* Improved presentation + clarity:
  * More details on gating
  * relationship to KV-cache and TTT
  * no clear relationship to $\epsilon$-greedy
  * more details on metrics, acronyms, probing results

(+ more in the rebuttals)

**Outcome**
- R2/3MFD and R3/7rrc have agreed and appreciated our rebuttal and asked us to incorporate the clarifications and new results into the camera ready version.
- R1/rguE did not raise specific weaknesses in spite of the 3/BR rating and did not yet interact.

We believe that the paper is now significantly stronger and thank the AC and the reviewers for their efforts, we know how taxing this can be, in particular during the summer period.

---

### Decision · Program_Chairs · 2025-09-17

**Decision:**

Accept (poster)

**Comment:**

In this paper, the authors propose Kinaema, a recurrent transformer sequence model designed for embodied agents that require long-horizon memory. The model maintains a distributed set of memory embeddings updated in O(1) time using a transformer block combined with GRU-style gating, avoiding the quadratic complexity of standard attention while maintaining high capacity. The method is evaluated on two new tasks: Mem-RPE (relative pose estimation from memory) and Mem-Nav (navigation to previously seen goals), showing improvements over recurrent baselines such as GRU, xLSTM, EMA, and MooG, particularly on long sequences.

Initial reviews were mixed (one accept, one borderline accept, one borderline reject). Strengths highlighted by the reviewers included the novelty of introducing an O(1) recurrent transformer memory (R-3MFD, R-7rrc), the introduction of Mem-RPE and Mem-Nav as valuable new benchmarks (R-3MFD, R-7rrc), and the comprehensive empirical comparisons across multiple strong baselines (R-3MFD, R-7rrc). R-rguE, while more negative, also acknowledged the motivation behind the architecture. On the other hand, weaknesses identified were the lack of coverage analysis and questions about whether memory was updated during navigation (R-3MFD), missing baselines such as state-space models (R-3MFD, R-7rrc), unclear distinction of Mem-Nav from ε-greedy exploration (R-7rrc), limited justification for training sequence lengths (R-3MFD), and general clarity issues around gating, notation, and figures (R-rguE, R-7rrc).

The authors provided a detailed rebuttal and discussion that addressed most of these concerns. They added new experiments analyzing SEEN vs. UNSEEN goals and extended training to 300M steps, which showed clearer Mem-Nav gains (addressing R-3MFD, R-7rrc). They made claridfications about Kinaema memory being updated during navigation, introduced state-space baselines (LRU and attempted Mamba), reporting significantly weaker performance for LRU and noting difficulties training Mamba (addressing R-3MFD, R-7rrc). They distinguished Mem-Nav from ε-greedy, provided an ε-greedy baseline, and clarified the PPO training setup (addressing R-7rrc). Finally, they committed to fixing the clarity issues noted by R-rguE and R-7rrc (notation, acronyms, tables/figures, gating explanation). Post-rebuttal, R-3MFD advocated acceptance and R-7rrc raised their score to borderline accept. R-rguE 's rating remained unchanged while the concerns were mainly focused on clairty.

The core contribution of introducing a scalable recurrent memory mechanism for embodied navigation is sound and well supported by additional experiments and clarifications. Given the novelty of the model, the new tasks introduced, the strong empirical foundation, and the reviewers mostly converging post-rebuttal, I recommend this paper be accepted, and highly recommend the authors incorporate the new results, baselines, and clarity improvements in the final version.